# Grounding line retreat and tide-modulated ocean channels at Moscow University and Totten Glacier ice shelves, East Antarctica

Tian Li[1,2], Geoffrey J. Dawson[1], Stephen J. Chuter[1], Jonathan L. Bamber[1,2]

[1]Bristol Glaciology Centre, School of Geographical Sciences, University of Bristol, Bristol, BS8 1SS, UK
[2]Department of Aerospace and Geodesy, Data Science in Earth Observation, Technical University of Munich, Ottobrunn, 85521, Germany

*Correspondence to*: Tian Li (tian.li@bristol.ac.uk) and Jonathan L. Bamber (j.bamber@bristol.ac.uk)

**Abstract.** The Totten and Moscow University glaciers, located in East Antarctica, contain 5.1 m sea-level equivalent of ice and have been losing mass over recent decades. Using ICESat-2 laser altimetry repeat track analysis and satellite radar
interferometry from Sentinel-1a/b SAR images, we mapped the grounding line (GL) locations of these two glaciers between 2017 and 2021. By comparing the 2017-2021 GL measurements with the historic GLs, we detected pervasive GL retreats along the ice plains at the glacier central trunk of Totten Glacier Ice Shelf (TGIS) and Moscow University Ice Shelf (MUIS). The GL retreated 3.51 ± 0.49 km at TGIS while it retreated 13.85 ± 0.08 km at MUIS since 1996. Using CryoSat-2 radar altimetry, we found that the observed GL retreats are coincident with high thinning rates, in addition to high ice velocities,
indicating a mass loss pattern dominated by ice dynamics. We also identified two tide-modulated ocean channels on Totten Glacier Eastern Ice Shelf (TGEIS) and Moscow University Western Ice Shelf (MUWIS), where the ocean channel widths are highly correlated with the differential tidal amplitudes. The opening of the MUWIS ocean channel connects the two previously separated TGIS and MUIS systems, which might open a pathway for the warm modified Circumpolar Deep Water to enter the main MUIS cavity and facilitate further GL retreat.

## 1 Introduction

The Antarctic Ice Sheet has contributed 7.6 ± 3.9 mm to global sea-level rise over the past 25 years (IMBIE team, 2018). Mass loss from East Antarctica is predominantly located in Wilkes Land and has been increasing since the 1970s (Rignot et al., 2019). Totten Glacier Ice Shelf (TGIS) and Moscow University Ice Shelf (MUIS) drain a large portion of the marine-based Aurora Subglacial Basin, which contains a sea-level rise equivalent of 5.1 m (Fig. 1a) comparable to the entire West
Antarctic Ice Sheet (Fretwell et al, 2013). Totten Glacier has the largest ice discharge in East Antarctica (71.4 ± 2.6 Gt yr$^{-1}$), with the basin it drains having the potential to raise sea level by 3.85 m (Rignot et al., 2019). With a smaller sea-level equivalent, the neighboring MUIS has an ice discharge of 47.0 ± 2.1 Gt yr$^{-1}$ (Rignot et al., 2019). Therefore, assessing the stability of these two glaciers is important in understanding the contribution of East Antarctica to future sea level rise.

Recent evidence has recognized the critical role played by ocean forcing in both the paleo large-scale retreat of Totten Glacier during the warm Pliocene epoch (Aitken et al., 2016), and modelled future rapid retreat at the Totten Glacier Eastern Ice Shelf (TGEIS) through the 21$^{st}$ century (Pelle et al., 2021). Observations suggest that warm modified Circumpolar Deep Water (mCDW) is widespread below 500 m water depth on the Sabrina Coast continental shelf (Fig.1) (Silvano et al., 2017). Driven by nearby activity in Dalton polynya, wind forcing or cyclonic eddies, the relatively warm salty water can encroach the sub-ice-shelf cavity through a deep trough (Nitsche et al., 2017), supplying sufficient heat to induce basal melting of the ice shelves (Rintoul et al., 2016; Greene et al., 2017; Hirano et al., 2021; Gwyther et al., 2014; Greenbaum et al., 2015). As a result, TGIS and MUIS have been undergoing high basal melting compared with other regions in East Antarctica (Pritchard et al., 2012; Depoorter et al., 2013; Adusumilli et al., 2020), including at their deep grounding lines (GLs) (Chuter and Bamber, 2015; Morlighem et al., 2020) - the location where the grounded ice first comes into contact with the ocean and becomes afloat.

Melting and thinning of ice shelves will reduce the buttressing of the grounded ice. If the GL is located on a retrograde bed slope, which is the case at TGIS and MUIS (Morlighem et al., 2020), it has the potential to trigger a positive feedback of increased ice discharge into the ocean through changes in ice velocity and GL positions (Joughin et al., 2014), a vulnerability that has been termed the Marine Ice Sheet Instability (MISI) (Schoof, 2007). The inland mass losses of TGIS and MUIS are concentrated in fast-flowing regions (Smith et al., 2020), meanwhile the DInSAR interferograms from the European Remote Sensing (ERS-1/2) radar satellites and COSMO-SkyMed (CSK) constellation showed that TGIS GL retreated 1 – 3 km between 1996 and 2013 (Li et al., 2015), indicating a change in ice flow dynamics in response to increased ice shelf thinning likely driven by ocean thermal forcing (Khazendar et al., 2013; Li et al., 2016). Despite the importance of this region, Li et al. (2015) has been the only study on GL migration of Totten Glacier from satellite observations due to the lack of available satellite data and limited spatial coverage. It is not known whether the TGIS GL has kept retreating at a similar pace as in 1996-2013, whether the GL of nearby MUIS is also retreating, or how the low-lying area between the TGIS and MUIS which is vulnerable to oceanic heat has been changing.

The high ice velocity of TGIS and MUIS limits the number of historic satellite observations of GL locations. Fast ice flow, together with katabatic wind and snowfall accumulation in this region, causes radar signal decorrelation for Differential Synthetic Aperture Radar Interferometry (DInSAR), making it difficult to discern the GL location (Goodwin, 1990; Li et al., 2015). Satellite laser altimetry can provide supplementary GL locations where DInSAR is not available, by measuring the tidally induced limits of ice flexure: the landward limit of tidal flexure Point F, the break-in-slope Point I$_b$ and the inshore limit of hydrostatic equilibrium Point H. These points constitute the grounding zone (GZ), which is the transition region between the fully grounded ice sheet and the freely floating ice shelf (Figure 1 in Fricker and Padman (2006)). The newly launched ICESat-2 satellite laser altimeter in 2018 can provide dense and high-resolution GZ locations across the Antarctic Ice Sheet (Li et al., 2022); however, its GZ measurements are not continuous in space. Therefore, combining DInSAR with satellite laser altimetry provides the optimal way to improve the GL coverage. In addition, studies show that tidally induced

GL changes may disguise the true GL retreat signal at a sub-annual scale, therefore this effect needs to be considered when investigating the long-term GL change (Milillo et al., 2017, 2022). The relatively short repeat cycle of DInSAR data (6-day for Sentinel-1a/b) enables a more detailed investigation of short-term GL migrations caused by ocean tides compared with laser altimetry (91-day for ICESat and ICESat-2). To study the GL changes of TGIS and MUIS at Wilkes Land, here we use a combination of ICESat-2 repeat track analysis from 2019-2021 and Sentinel-1a/b DInSAR data during 2017-2021 to map the GL locations and evaluate the short-term GL variability. We then evaluate the long-term GL migrations by comparing with the historic DInSAR-derived GLs from the MEaSUREs project (Rignot et al., 2016, 2011), the ice sheet elevation change rates calculated from CryoSat-2 radar altimetry swath data (Gourmelen et al., 2018) and DInSAR-derived ice velocities (Mouginot et al., 2017b).

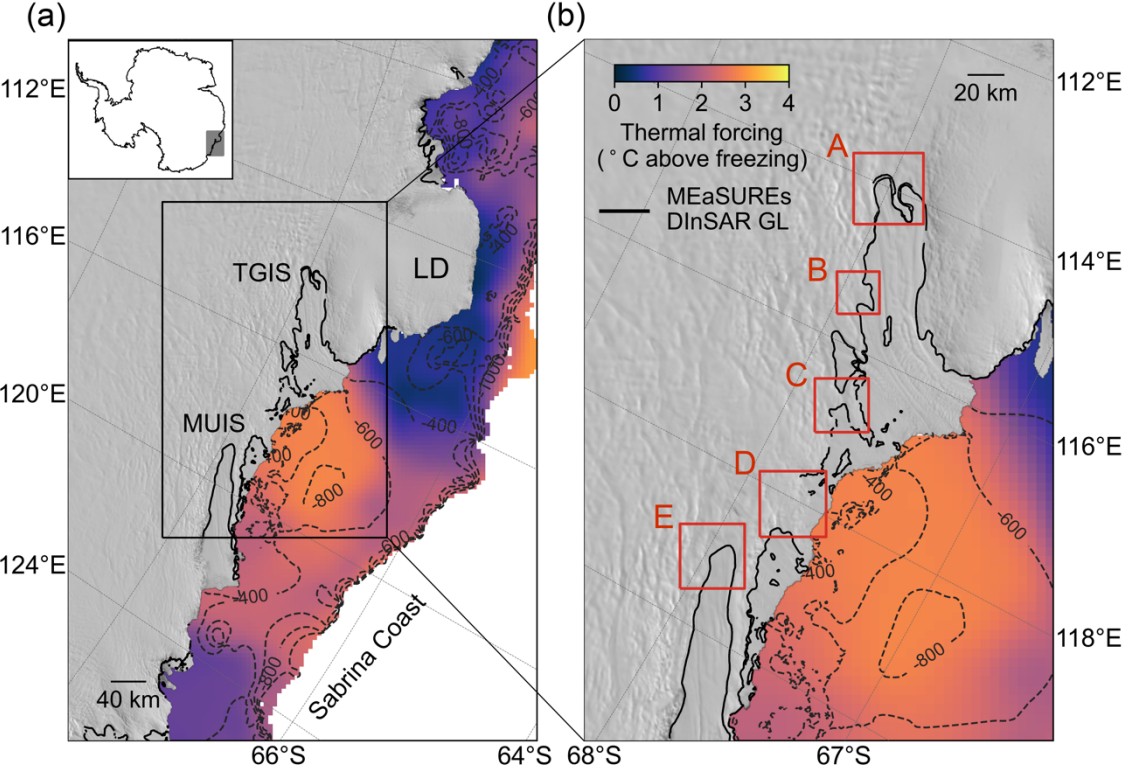

Figure 1. a) Totten Glacier Ice Shelf (TGIS), Moscow University Ice Shelf (MUIS) and Law Dome (LD) in Wilkes Land overlaid with the MODIS mosaic of Antarctica (Scambos et al., 2007), and the maximum ocean thermal forcing (temperature above the in situ freezing point) between water depth 200 − 800 m (Adusumilli et al., 2020) on Sabrina Coast. The MEaSUREs Differential Synthetic Aperture Radar Interferometry (DInSAR) grounding lines (GLs) are shown as black lines (Rignot et al., 2011, 2016), the BedMachine bed elevation (Morlighem et al., 2020) contours at -400, -600, -800 and -1000 m of the seafloor are shown as dashed dark grey lines. b) Zoomed-in map of the TGIS and MUIS outlined as the black box in a); A, B, C, D, E red boxes denote five studied regions including TGIS, Totten Glacier East Branch (TGEB), Totten Glacier Eastern Ice Shelf (TGEIS), Moscow University Western Ice Shelf (MUWIS), and MUIS, respectively.

## 2 Data and Methods

### 2.1 Grounding line mapping from Sentinel-1a/b DInSAR interferograms

The single look complex (SLC) synthetic aperture radar (SAR) images in wide swath mode from both Sentinel-1a/b satellites with a 6-day repeat cycle were used to construct DInSAR interferograms between July 2017 and September 2021. These interferograms were used to derive the GL locations of the Moscow University and Totten Glacier ice shelves in Wilkes Land. The data were processed using the GMTSAR software (Sandwell et al., 2011) and the surface topography was removed using the 8 m resolution Reference Elevation Model of Antarctica (REMA) Digital Elevation Model (DEM) (Howat et al., 2019). The DInSAR was generated by differencing two consecutive 6-day SAR interferograms. This differencing removes signals such as time-invariant velocity, to identify ice flow deformation signals caused by ocean tides at the GZ. We processed 139 DInSAR interferograms and manually delineated GLs according to the guidelines in Brancato et al. (2020), by counting the number of fringes between the grounded ice and the floating ice and choosing the most upstream interferometric fringe which represents the furthest point inland influenced by ocean tides. Due to poor radar signal coherence in the fast-flowing regions, we only managed to obtain three interferograms containing GL locations at the MUIS main glacier trunk and failed to derive any useful interferograms at the TGIS main glacier trunk. However, we were able to map the GLs at Totten Glacier East Branch (TGEB, box B in Figure 1b), Moscow University Western Ice Shelf (MUWIS, box D in Figure 1b) and TGEIS (box C in Figure 1b), with the numbers of usable DInSAR interferograms in each region are 11, 7 and 14, respectively. On average, the GL mapping precision is 2 pixels, equivalent to a mapping error of approximately 90 m.

### 2.2 Grounding zone mapping from ICESat

Brunt et al. (2010a) produced a GZ product for Antarctic ice shelves using ICESat laser altimetry datasets between 2003 and 2009. There only existed two ICESat-derived Point F locations at the MUIS main glacier trunk from the Brunt et al. (2010a) product and they were not able to map any GZ points at the main glacier trunk of TGIS. While this dataset can be used as a baseline in comparing GL change, the GZ is difficult to map in fast-flowing ice streams and can be prone to errors (Dawson and Bamber, 2017). Therefore, we reprocessed the track 323 of ICESat L2 Global Antarctic and Greenland Ice Sheet Altimetry Data product GLAH12, version 34 (Zwally et al., 2014), to derive GZ information at MUIS. The newly-mapped ICESat-derived Point F locations at MUIS were also used to validate the historic DInSAR-derived GLs from the MEaSUREs project (Rignot et al., 2011, 2016).

The ICESat satellite was in operation between 13[th] January 2003 and 11[th] October 2009. We used the ICESat campaigns between October 2003 and October 2009 with a repeat cycle of 91 days. The release 34 dataset has been corrected for the Gaussian-Centroid (G-C) offset (Borsa et al., 2014). We applied the saturation correction to reduce the laser-energy-related

impacts on elevations (Zwally et al., 2014). To remove the elevation biases caused by long-period variations in range measurements over different mission campaigns, we applied the inter-campaign bias (ICB) corrections determined from Hofton et al. (2013) to each campaign. The ocean tide and ocean loading tide corrections were removed from the elevations as the GZ mapping requires sampling vertical elevation changes at different tidal amplitudes (Zwally et al., 2014). We removed the data affected by clouds with the Gain Value used for Received Pulse i_gval_rcv > 200 (Bamber et al., 2009a),

and removed the repeat tracks with an across-track distance larger than 100 m to eliminate the across-track slope-induced errors in elevation anomaly calculation (Brunt et al., 2010b). To calculate the elevation changes caused by ocean tides at different repeat cycles, the elevations of each repeat track were interpolated to the reference orbit track with an along-track distance of 200 m. The elevation anomalies were calculated by subtracting the mean elevation of all repeat tracks on this reference track from the elevation of each repeat track profile. We obtained the elevation anomalies of track 323 across the

main glacier trunk of MUIS and manually identified Point F locations (red vertical dashed lines in Figures S1a-b, red crosses in Figure S1c).

## 2.3 Grounding zone mapping from ICESat-2

The Advanced Topographic Laser Altimeter System (ATLAS) onboard ICESat-2 measures the ice surface elevation using six beams in three beam pairs with a 91-day repeat cycle (Markus et al., 2017). The across-track separation between each

beam pair is about 3.3 km with a pair spacing of about 90 m. The beam pair setting allows an instantaneous determination of local across-track slopes (Smith et al., 2019). In this study, the Land Ice Along-Track Height Product ATL06 version 4 data (Scheick, 2019; Smith et al., 2021) from 30[th] March 2019 to 1[st] June 2021 were used, including 9 repeat cycles (3-11), among which cycles 4 and 11 are not complete. Our approach of processing ATL06 data and estimating the GZ features including Point F and Point H, closely follows the methodology described in Li et al. (2020, 2022). Here we briefly review

the key features of this method.

The GZ mapping is based on detecting the vertical movement of floating ice caused by ocean tides, therefore the ocean tide correction was not applied to the ICESat-2 ATL06 elevation, and the elevations were 're-tided' by removing the ocean-loading tide correction (Smith et al., 2021). Ground tracks of four Reference Ground Tracks (RGTs) 163, 841, 1108, 1283 were used at MUIS, while ground tracks of RGTs 87, 262, 1032, 1207 were used at TGIS. Each RGT has six ground tracks,

they were further categorized into nine distinct repeat-track data groups, including six single-beam repeat-track data groups and three beam-pair repeat-track data groups (Figures 4a, b in Li et al. (2020)). For each data group, a nominal reference track was calculated, as well as a reference GL, which was the intersection between the nominal reference track and the composite GL described in Li et al. (2022).

For MUIS, the reference GL was defined roughly based on the break-in-slope from the REMA DEM (Howat et al., 2019)

(white dashed line in Figure S2a) to take into account different orientations of ICESat-2 tracks across the GZ and possible

GL retreat during the past two decades. This ensures that no GL retreat is omitted in our study caused by the uncertainty in predefined reference GL locations on MUIS. In addition, the elevations of each repeat track in the beam-pair repeat-track data group were corrected for the across-track slope on the nominal reference track (Equations 1, 2 in Li et al. (2020)). A set of elevation anomalies was then calculated by differencing each individual repeat track elevation profile and a reference elevation profile, which is defined as the average of elevations for each repeat track along the nominal reference track (Li et al., 2020, 2022).

The estimation of GZ features is based on extracting the transition points from the mean absolute elevation anomaly (MAEA), which is defined as the average of the absolute value of all elevation anomaly profiles (Li et al., 2020). The limit of tidal flexure, Point F, is identified as the point where the elevation anomaly of each repeat track exceeds a noise threshold (Fricker et al., 2009; Brunt et al., 2010b, 2011), and the point where the gradient of the MAEA first increases from zero and the second derivative of the MAEA reaches its positive peak (Li et al., 2020). The point of hydrostatic equilibrium, Point H, is identified as the location where the elevation anomaly of each repeat track reaches its maximum and becomes consistent with the predictions from CATS2008 ocean tide model, which is an update to the model described by Padman et al. (2002), and the transition point where the gradient of the MAEA finally decreases to zero and the second derivative of the MAEA reaches its negative peak (Li et al., 2020).

To select the correct transition points from the second derivative of the MAEA curve as Points F and H, an error function was fitted to the MAEA as a guide to find the Point H. In addition, a three-segment piecewise function was fitted to the landward part of Point H of the MAEA profile to find Point F (Li et al., 2022). The closest positive peak of the second derivative of this piecewise function to the reference GL was taken as a guide to find Point F. All results are visually inspected due to the complex nature of the GZs as a final step. In this study, time stamp of the ICESat-2-derived GZ feature was taken to be 2020, which is the median of the ICESat-2 data period used in this study.

## 2.4 Grounding line migration distances

The GL migration distances were measured at MUIS, TGIS, TGEB and MUWIS along ice flowlines by comparing the present-day GLs mapped in our study with the 1996 ERS-1/2 DInSAR-derived GL from the MEaSUREs project (Rignot et al., 2011, 2016). For MUIS and TGIS, the GL migration distances were calculated as the along-flowline separations between the selected ICESat-2-derived Point F locations and the 1996 MEaSUREs GL (Fig. S2). At these two regions, the ice flowlines were selected based on two criteria: 1) passing through the ICESat-2-derived Point F locations, and 2) having intersections with the 1996 MEaSUREs GL. In addition, the selected ice flowlines should be near parallel to each other and close in space to reduce uncertainties in GL migration rates due to discrepancies in spatial distribution. At MUIS, the GL migration distances were also measured by comparing the Sentinel-1a/b DInSAR-derived GLs with the 1996 MEaSUREs GL along an ice flowline, which should locate at the glacier central trunk and has intersections with both Sentinel-1a/b

DInSAR GLs and the MEaSUREs 1996 GL. The GL migration distance measurements at TGEB and MUWIS follow the same approach by comparing the Sentinel-1a/b DInSAR-derived GLs with the 1996 MEaSUREs GL.

## 2.5 CryoSat-2 elevation change rates from swath altimetry

The CryoSat-2 Swath mode thematic point product from the European Space Agency (ESA) CryoTEMPO-EOLIS project (Gourmelen et al., 2018) was used to determine the mean rate of ice sheet elevation change ($\Delta h/\Delta t$) for the period July 2010 – December 2019 (Chuter et al., 2022). Swath processing increases the number of elevation observations by about two orders of magnitude in comparison to using the previous standard point of closest approach (POCA) CryoSat-2 products. Before determining elevation trends, in addition to the quality filtering used in the creation of the CryoTEMPO-EOLIS

product, the potential for outliers was further reduced by removing all observations with an uncertainty score > 5m. $\Delta h/\Delta t$ is determined through a linear plane fit procedure at a regular 5 km grid posting across the ice sheet using Eq. (1):

$$z(x, y, t) = \bar{z} + a_1 x + a_2 y + a_3 xy + a_4 t + \varepsilon, \quad (1)$$

Where z is the surface elevation, x and y are the spatial coordinates, t is time and $\varepsilon$ is the observation noise. The coefficients $a_1$, $a_2$ and $a_3$ resolve for variations in ice sheet surface topography within each grid cell. Due to swath elevation data

typically having a larger noise level than conventional POCA products (Gray et al., 2017), an iterated re-weighted least squares approach was used, with model fit residuals > 2$\sigma$ removed after each iteration until no residual outliers remained.

Additionally, grid cells containing potentially poorly fitted planes were removed if any of the following conditions were met: an absolute $\Delta h/\Delta t$ > 15 m yr$^{-1}$, surface slope > 3° and a plane fit uncertainty > 1 m yr$^{-1}$. In addition, a 50 km kernel Median Absolute Deviation (MAD) filter was used to remove $\Delta h/\Delta t$ values which have a value > 2 MAD. Finally, a 15 km median

filter was applied to reduce any artifacts of the gridded values.

## 2.6 Height above hydrostatic equilibrium

The height above hydrostatic equilibrium $h_f$ was calculated using Eq. (2),

$$h_f = h - H \frac{(\rho_w - \rho_i)}{\rho_w}, \quad (2)$$

where the ice-equivalent ice thickness $H$ and surface elevation $h$ referenced to mean sea level available from the

195 BedMachine Antarctica dataset were used (Morlighem et al., 2020; Morlighem, 2020). The sea water density $\rho_w$ and ice density $\rho_i$ are 1,027 kg m$^{-3}$ and 917 kg m$^{-3}$, respectively. The nominal error is 11 m based on the uncertainty of 2 m in surface elevation and 100 m in bed topography.

### 2.7 Other datasets

In addition to the Sentinel-1a/b DInSAR-derived GLs and the ICESat and ICESat-2-derived GZs mapped in this study, we used the historic GLs available from the MEaSUREs project which were mapped from ERS-1/2 in 1996 and CSK constellation in 2013 as reference GLs to analyse the GL migration (Rignot et al., 2016, 2011). The deep-learning-based Antarctic GLs produced from Sentinel-1a/b SAR images in 2018 by Mohajerani et al. (2021) were also used as a reference for comparing GL migrations in this study.

Two different tide models CATS2008 (Padman et al., 2002) and FES2014 (Lyard et al., 2021) were used to calculate the differential tidal amplitudes δh at each DInSAR GL measurement using Eq. (3),

$$\delta h = (h4 - h3) - (h2 - h1), \quad (3)$$

where h1, h2, h3, h4 are the modelled tidal amplitudes at the acquisition time of each SAR image pass used in the DInSAR interferogram. The tidal range for the DInSAR GL measurement is calculated as the difference between the maximum and minimum tidal amplitudes.

To study the ice velocity changes at the GZs, the MEaSUREs annual ice velocity maps with a 1 km grid spacing, version 1, between 2005 and 2016 were used (Mouginot et al., 2017a, b). This dataset was derived from a combination of InSAR-based analysis of multiple SAR data and the feature tracking of Landsat-8 images. For ice velocities between 2017 and 2018, the ITS_LIVE annual ice velocity datasets derived from Landsat optical images using features tracking were used (Gardner et al., 2018, 2019).

## 3 Results

### 3.1 Grounding line migration in fast-flowing areas

#### 3.1.1 Moscow University Ice Shelf

The only publicly available historic measurements of Point F with full spatial coverage at the main glacier trunk of MUIS come from the 1996 MEaSUREs DInSAR-derived GL (Rignot et al., 2011, 2016). However, this interferogram cannot be replicated in this study due to the lack of available Level 1 ERS-1/2 SAR Simple Look Complex (SLC) data in 1996 in this region. Nevertheless, the elevation anomaly analysis of ICESat repeat tracks along RGT 323 (Fig. S1) between June 2004 and November 2005 in our study matches well with the 1996 GL. It identifies an erroneous geolocation of Brunt et al. (2010a)'s ICESat-derived Point F at the eastern flank of the MUIS GZ (purple cross in Figure S1c). The proximity between the 1996 GL and newly derived ICESat Point F locations along RGT 323 gives us confidence that the 1996 GL is accurate and reliable, and we use, therefore, the 1996 GL as a baseline to evaluate the GL migration rates in this region. However, the

RGT 323 is the only ICESat ground track available at the MUIS GZ and it is not situated on the main glacier trunk. It is important to note that the MEaSUREs GL dataset in 1996 only provides one GL measurement in areas not covered by the ICESat repeat tracks, which does not allow us to characterize the historic tidally induced GL variability.

Three Sentinel-1a/b DInSAR interferograms with a 6-day repeat cycle (Figs. 2d-f) show a rapid short-term GL migration of 230    2.53 ± 0.13 km in just six days between 23$^{rd}$ and 29$^{th}$ April 2021  (Table S1). A high positive correlation is found between the GL migration distance and the differential tidal amplitude δh with an R-squared value of 0.91 (p-value > 0.1 given we only have three samples) (Fig. 3), indicating this short-term GL migration is likely driven by ocean tides. This symmetric GL migration over a 12-day period, however, contradicts the nonlinear and asymmetric GL migration over a tidal cycle simulated by Tsai and Gudmundsson (2015) using an elastic fracture model. The contradiction can be possibly attributed to 235    the fact that we only have limited three DInSAR GL measurements over this region and the bedrock elevation at MUIS GZ is poorly known (Morlighem et al., 2020; Morlighem, 2020) which may influence the model output from Tsai and Gudmundsson (2015). Compared with the 1996 GL, the furthest GL migration inland along the ice flowline among these three interferograms is 5.67 ± 0.13 km. The mean GL retreat of these three DInSAR interferograms relative to the 1996 GL is 4.22 km with a standard deviation of 1.07 km. The variation in GL location caused by short-term ocean tides is shorter 240    than the smallest GL migration of 3.14 km on 23$^{rd}$ April 2021 since 1996. In addition, assuming that the tide-induced GL fluctuation is constant over time, the modelled differential tidal amplitude δh of -11.22 cm from CATS2008 for the historic 1996 MEaSUREs GL (Table S2) is equivalent to a GL migration distance of 4.35 km in April 2021 since 1996 (vertical orange dotted line in Figure 3). This indicates that the observed GL migration from DInSAR includes a long-term trend.

To supplement the limited Sentinel-1a/b DInSAR measurements, the elevation anomalies of repeat tracks along four 245    different RGTs of ICESat-2 laser altimetry data (163, 841, 1108 and 1283) across the same region were calculated to derive Point F locations (maroon dots in Figure 2a and vertical red dashed lines in Figure 4). Ten ICESat-2-derived Point F locations at the southernmost portion of the MUIS GZ were selected to calculate the GL migrations along the ice flowlines (Fig. S2a) by comparing with the 1996 DInSAR GL. The average 1996-2020 GL retreat distance at the western flank of MUIS GZ is 13.85 km, while it is 9.37 km at the eastern flank of the MUIS GZ (Table S3). Large GL deviations (~5.9 km 250    along ice flowline, Fig. 2a) exist between the ICESat-2-derived Point F at Track 1108 GT2R (8 March – 7 June 2020) (Fig. 4a) and the Sentinel-1a/b DInSAR GL on 29$^{th}$ April 2021 (Fig. 2e). One possible explanation for this deviation is short-term GL migrations caused by different ocean tidal amplitudes that are not fully captured in our study.

The Sentinel-1a/b DInSAR interferograms in June 2018 show the existence of an oval-shaped feature at the southeast flank of the MUIS GZ (Figs. 2b, c), which was previously detected as a GL in Mohajerani et al. (2021). This oval-shaped feature 255    is not constant over time on the interferogram, disappearing in September 2021 with another similar oval-shaped feature appearing in the north (Fig. 2g). The repeat tracks of three ICESat-2 ground tracks RGT 163 GT2L (Fig. S3), RGT 338 GT2R (Fig. S4) and RGT 605 GT3L (Fig. S5), indicate that the two unusual lobate shaped features are probably subglacial

lakes with high hydrostatic potential (Fig. S6). The ICESat-2 elevation anomaly time series of tracks 163 and 338 show gradual surface uplift between 20 April 2019 (cycle 3) and 14 October 2021 (cycle 13) on the southern lake, with a height

change up to 10 m (Fig. S4). In the meantime, track 605 shows a similar uplifting event between 3 May 2021 (cycle 11) and 1 November 2021 (cycle 13) on the northern lake, and the height change is around 2 m during this period (Fig. S5).

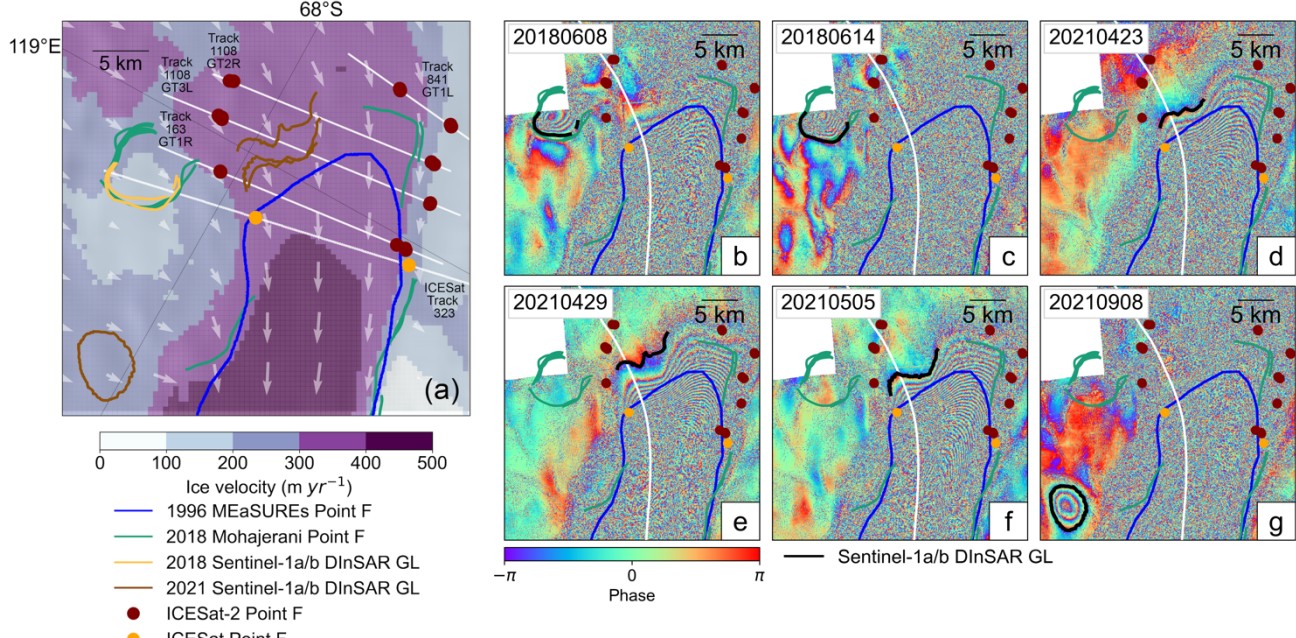

**Figure 2. a) Grounding line (GL) distributions at the main glacier trunk of Moscow University Ice Shelf (MUIS) overlaid with the ice surface velocity magnitudes and ice flow directions (white arrows) (Rignot et al., 2017). The ICESat-2-derived inland limit of**
**tidal flexure (Point F) locations are shown as maroon dots, and the ICESat-derived Points F locations from Figure S1 are shown as orange dots. b-g) Sentinel-1a/b DInSAR interferograms of MUIS between 2018 and 2021, the ice flowline is shown as the white solid line. In all subplots, the 1996 MEaSUREs DInSAR GLs (Rignot et al., 2011, 2016) and the 2018 DInSAR GLs (Mohajerani et al., 2021) are shown as blue solid lines and green solid lines, respectively. The GLs delineated from Sentinel-1a/b DInSAR interferograms in our study are shown as yellow (2018) and brown (2021) solid lines in subplot a, and black solid lines in subplots**
**b-g.**

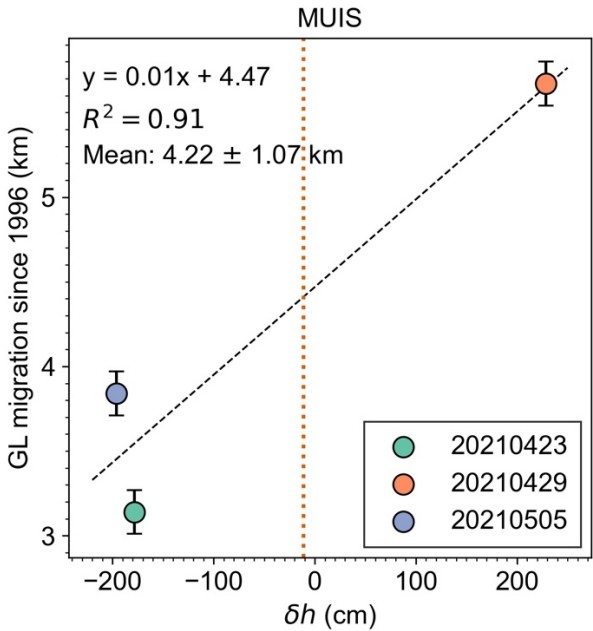

**Figure 3.** Correlation between the differential tidal predictions from the CATS2008 tidal model at the acquisition dates of each interferogram and the grounding line (GL) migrations since 1996 for the three Sentinel-1a/b DInSAR interferograms in Figures 2d-f (Table S1). The GL migration is measured along the ice flowline (white solid line in Figures 2d-f). Positive GL migration values indicate that the GL retreated inland with respect to the reference point, which is the intersection between the 1996 DInSAR-derived GL (Rignot et al., 2011, 2016) and the ice flowline. The vertical orange dotted line denotes the differential tidal amplitude calculated from the CATS2008 tidal model at the time of four satellite image passes used in the MEaSUREs 1996 ERS-1/2 DInSAR GL measurement (Rignot et al., 2011, 2016) (Table S2). The geolocation of tidal prediction reference point is shown in Figure S7.


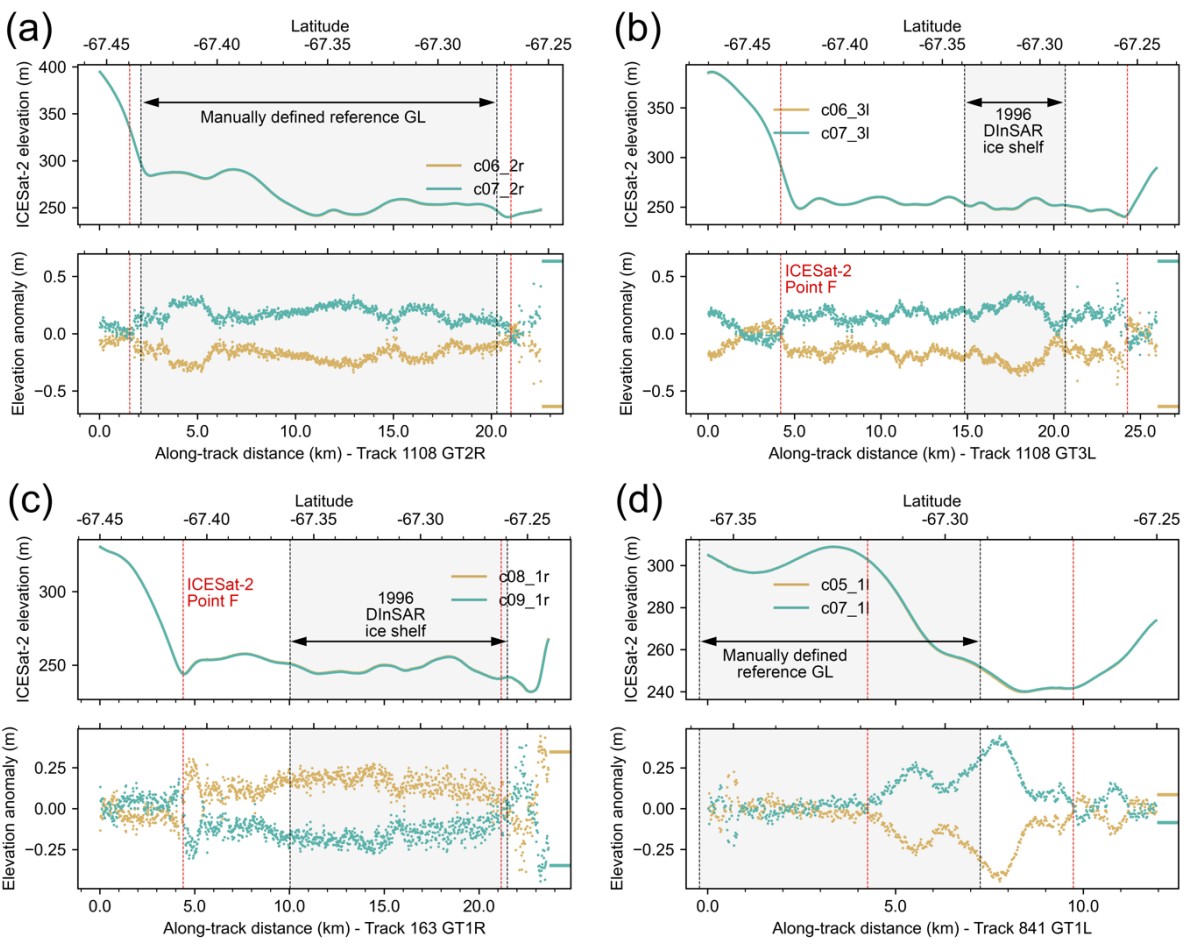

**Figure 4. The ICESat-2 elevation anomaly analysis for different elevation profiles on the main glacier trunk of Moscow University Ice Shelf (MUIS) shown in Figure 2a, including track 1108 GT2R (a), track 1108 GT3L (b), track 163 GT1R (c) and track 841 GT1L (d). For all subplots, the upper panel shows the along-track ICESat-2 surface elevation and the lower panel shows the**
**elevation anomalies, the ICESat-2-derived Point F location is shown as the vertical red dashed line, the zero mean ocean tidal amplitudes predicated from the CATS2008 tidal model at the location of -67.0714° S, 120.6249° E are shown as the horizontal line segments on the right side of the bottom panel of each plot.**

### 3.1.2 Totten Glacier Ice Shelf

High ice velocity at the fast-flowing TGIS main glacier trunk and the 6-day repeat cycle for Sentinel-1a/b satellites cause a
loss of phase coherence in all Sentinel-1a/b DInSAR interferograms. Therefore, no GLs can be reliably detected from Sentinel-1 DInSAR measurements at the TGIS main glacier trunk in our study. With ICESat-2 repeat tracks, however, we were able to identify GZ locations (red dots in Figure 5a) at the southern lobe of TGIS (box A in Figure 1b). Four examples of the ICESat-2 repeat track analysis for track 1207 GT2R, track 1207 GT3L, track 1032 GT1L and track 1032 GT2R are

shown in Figure S8. The historic GLs in 1996 and 2013 in this region were mapped with high confidence according to
Figure 2 in Li et al. (2015), therefore they were used as a reference in calculating GL migration rates with ICESat-2-derived Point F locations along the ice flowlines shown in Figure S2b.

The GL migration and retreat rates at the southern lobe of TGIS between 1996 and 2020 are listed in Table S4. During 2013-2020, the GL retreated $0.75 - 2.33$ km which is equivalent to an average retreat rate of $0.11 - 0.33$ km yr$^{-1}$. In comparison, the GL retreated $1.93 - 2.17$ km during 1996-2013, equivalent to an average GL retreat rate of $0.11 - 0.13$ km yr$^{-1}$. In
addition to calculating GL migration rates, we investigated the short-term variation in GLs modulated by ocean tides (Fig. 5b). We directly calculated the tidal range as the difference between the maximum and minimum ICESat-2 elevation anomalies at Point H for each GZ measurement. Given that the chosen ICESat-2-derived Points F are close by and the corresponding ice flowlines are near parallel at the GZ (Fig. S2b), deviations in GL migration caused by spatial discrepancy in ice flowline locations should be small. A high correlation between GL migration and tidal range with an R-squared value
of 0.71 (p-value < 0.05) was found. The average GL migration is 3.51 km between $1996 - 2020$, and the standard deviation of 0.49 km was taken as the spatial variation caused by the short-term tidal amplitude changes during this period. The CATS2008 modelled tidal range for the 1996 MEaSUREs ERS-1/2 DInSAR GL is 48.68 cm (Table S2), the GL migration distance since 1996 at this tidal range is 2.69 km by assuming a constant tide-induced GL fluctuation at TGIS according to Figure 5b.


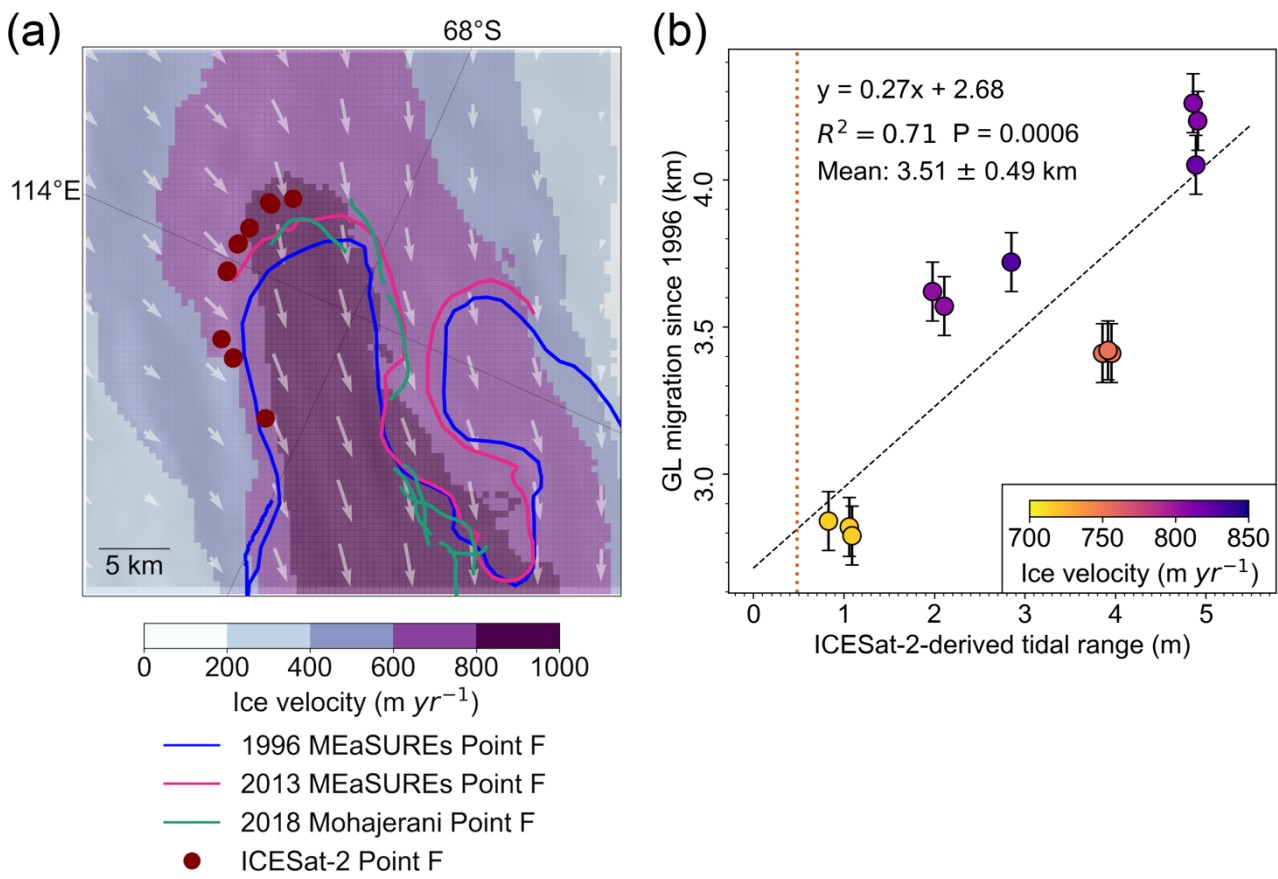

**Figure 5. a) Grounding line (GL) distributions at the main glacier trunk of the Totten Glacier Ice Shelf (TGIS) overlaid with the ice surface velocity magnitudes (Rignot et al., 2017) and ice flow directions (white arrows). The ICESat-2-derived Point F locations are shown as maroon dots. The 1996 and 2013 DInSAR-derived GLs (Rignot et al., 2011, 2016) are shown as blue and pink solid lines, respectively. The 2018 DInSAR-derived GLs (Mohajerani et al., 2021) are shown as green solid lines. b) The ICESat-2-derived tidal range for each Point F at the selected ice flowlines in Figure S2b and the GL migration distance between DInSAR-derived GL in 1996 (Rignot et al., 2011, 2016) and ICESat-2-derived Point F along ice flowlines (Table S4). The vertical orange dotted line denotes the tidal range calculated from the CATS2008 tidal model at the time of the satellite passes used in MEaSUREs 1996 ERS-1/2 DInSAR grounding line measurement (Rignot et al., 2011, 2016) (Table S2).**

### 3.1.3 Totten Glacier East Branch

Located at the eastern flank of TGIS, Totten Glacier East Branch (TGEB, box B in Figure 1b) is a relatively fast-flowing ice stream with an ice velocity of about 200 m yr⁻¹. Eleven Sentinel-1a/b DInSAR interferograms between 2018 and 2021 in Figure 6 show that the GL has been retreating since 1996 (Table S5). The GL retreat distances along the same ice flowline (white solid line in Figure 6a) vary between 4.64 – 6.96 km. In addition, we calculated a mean GL retreat distance to be 5.95 km from DInSAR interferograms, and the standard deviation is 0.75 km. In contrast to MUIS and TGIS main glacier trunks

(Figs. 3 and 5b), the tidal ranges calculated from the CATS2008 tidal model and the GL retreat distances are not correlated (R-squared value is 0.16, p-value > 0.1) (Fig. 7). Although it is unclear if this lack of correlation is caused by the inaccurate tidal model predictions in this region, different fringe numbers of each interferogram (Fig. 6) indicate a variation in ocean tidal amplitudes which can change the GL shape at the chosen ice flowline as well as the final GL migration distances. In

addition, the fact that the magnitude of short-term GL migration (0.75 km) is much smaller than the mean GL migration distance since 1996 indicates that the observed landward GL migrations are mainly from long-term GL retreat.

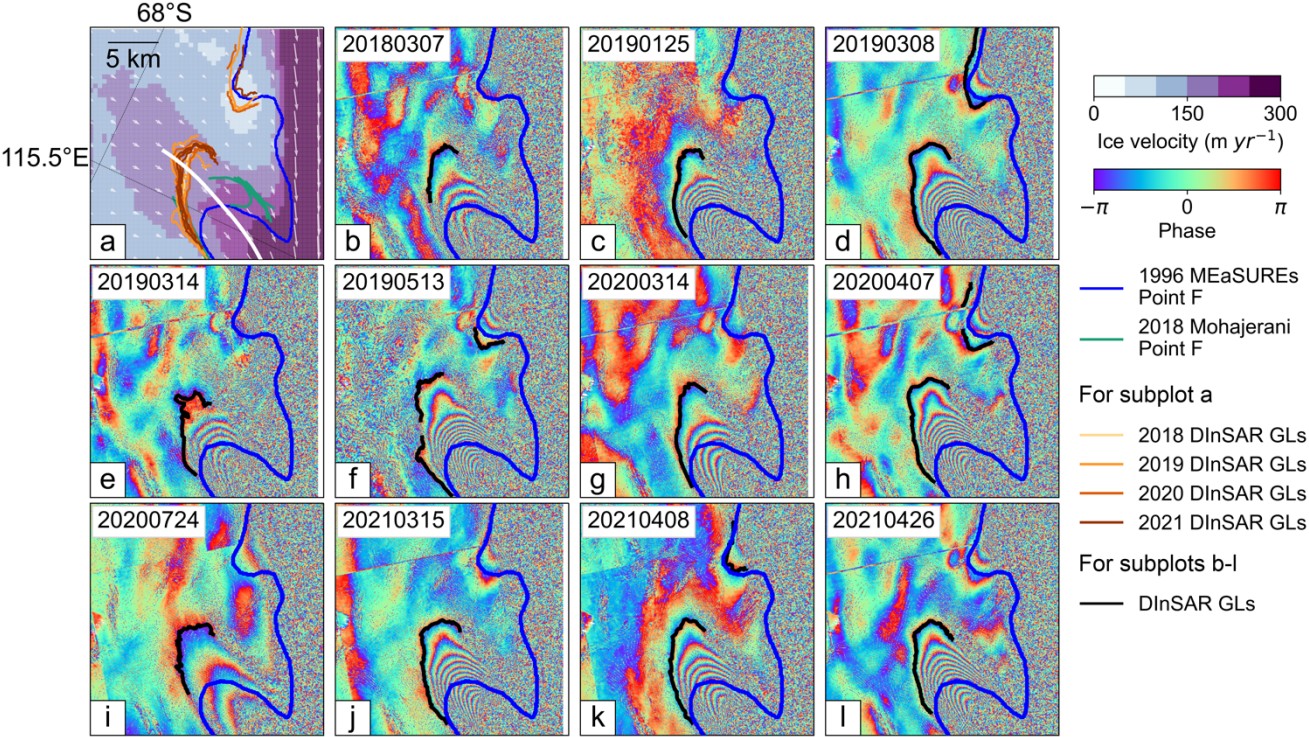

**Figure 6. a) Distribution of different grounding line (GL) products at Totten Glacier East Branch (TGEB) overlaid with the ice surface velocity magnitudes** (Rignot et al., 2017) **and ice flow directions (white arrows). The Sentinel-1a/b DInSAR GLs mapped in**
**this study are color-coded by year. b-l) GLs (black solid line) delineated using Sentinel-1a/b DInSAR interferograms between 2018 and 2021. In all subplots, the 1996 MEaSUREs DInSAR-derived GLs (Rignot et al., 2011, 2016) are shown as blue solid lines. The ice flowline used to measure the GL migration rates is shown as the white solid line and the 2018 DInSAR-derived GLs (Mohajerani et al., 2021) are shown as the green solid lines in subplot a).**

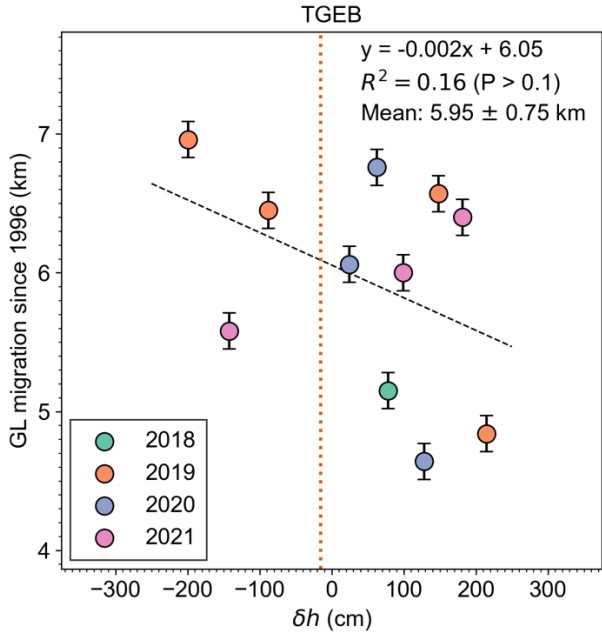


**Figure 7. Comparison between the grounding line (GL) migration and differential tidal predictions δh from the CATS2008 tidal model (Padman et al., 2002), at acquisition dates of each DInSAR interferogram at Totten Glacier East Branch (TGEB). Positive GL migration values indicate that the GL retreated inland with respect to the reference point, which is the intersection between the 1996 DInSAR-derived GL (Rignot et al., 2011, 2016) and the ice flowline in Figure 6a. The vertical orange dotted line denotes**
**the differential tidal predication calculated from the CATS2008 tidal model at the time of the satellite passes used in MEaSUREs 1996 ERS-1/2 DInSAR GL measurement (Rignot et al., 2011, 2016) (Table S2). The geolocation of tidal prediction reference point is shown in Figure S7.**

## 3.2 Tide-modulated ocean channels

Two ocean tide modulated channels at the low-lying areas of TGIS and MUIS - one located at MUWIS (Fig. 8, box D in Fig.
1b) and the other located at TGEIS (Fig. 9, box C in Fig. 1b), were identified from the Sentinel-1a/b DInSAR interferograms using the same method in Section 2.1. On 17[th] April 2021, the MUWIS channel was closed due to low tidal changes (Fig. 8e, Table S6). Then its width expanded to 2.84 ± 0.13 km on 29[th] April (Fig. 8g) in just 12 days (Table S6). The channel width is positively correlated with the absolute differential tidal amplitude |δh| (Fig. 10a) with an R-squared value of 0.48 (p-value < 0.1). Due to a loss of coherence in the Sentinel-1a/b DInSAR interferograms, we are unable to map the GL locations along
most of the southern region of MUWIS (Fig. 8). Despite this, the opening of this ocean channel indirectly confirms that the GL has retreated further inland to the channel entry. This is also supported by the DInSAR GL observation on 5[th] May 2021 in Figure 8h (black solid line) on the ice shelf southern GZ. The GL migration distance along an ice flowline (white solid line in Figure 8a) between 5[th] May 2021 and 1996 is 11.95 ± 0.13 km, equivalent to an average retreat rate of 0.48 km yr[-1].

As this is the only Sentinel-1a/b DInSAR interferogram that contains a GL signal at the southern flank of MUWIS in this study, the possibility of short-term GL fluctuations caused by ocean tides cannot be ruled out. However, the ESA CCI DInSAR GL location in 1996 (ESA, 2017) (yellow line in Fig. S9d) mapped from ERS-1/2 SAR images matches well with the MEaSUREs GL in this region (blue line in Fig. S9d). This suggests that the large GL migration in this region is dominated by a long-term trend and not the short-term tidal variability.

Another example of a tide-modulated ocean channel is located at TGEIS. Previous research using radar sounding data and laser altimetry datasets discovered the existence of an ocean trough at TGEIS (Greenbaum et al., 2015). This enables an easier entry of warm mCDW into the main TGIS cavity. Similarly, the Sentinel-1a/b DInSAR interferograms in our study show the existence of this channel (Fig. 9). From these interferograms, we observe that the channel width is not constant, but correlated with the absolute differential tidal amplitude |δh| with an R-squared value of 0.66 (p-value < 0.05) (Fig. 10b). For example, when differential tidal amplitudes were low, the channel was completely closed (Figs. 9b, e, j and m). When the differential tidal amplitudes increased from 17[th] April to 29[th] April 2021, the channel reopened, and its width increased to 2.89 ± 0.13 km (Table S7). This correlation is higher than the MUWIS ocean channel. The most likely causes of the lower correlation coefficient at MUWIS are that we have fewer DInSAR observations in this region. Additionally, the narrower ocean trough may lead to ice not fully reaching hydrostatic equilibrium, therefore bending forces may also influence the response of the ice to ocean tide variations. In Greenbaum et al. (2015), they claimed it is difficult to determine whether the TGEIS channel formed around 2010, or before 1996 given the possibility that ERS-1/2 DInSAR interferogram may fail to capture the correct tidal signal due to signal coherence issues. The results here support a third possibility: that the 1996 ERS-1/2 DInSAR-derived GL is accurate but the ERS satellite passes happened to be at low tides when the channel was closed. We calculated the |δh| at the acquisition times of the ERS-1/2 SAR images used in the MEaSUREs 1996 DInSAR-derived GL over this cavity (Rignot et al., 2016), which is 8.6 cm from CATS2008 (vertical orange dotted line in Figure 10b) or 9.77 cm from the FES2014 tidal model (Table S2). According to the linear regression in Figure 10b, the channel width should be close to zero at this differential tidal amplitude.

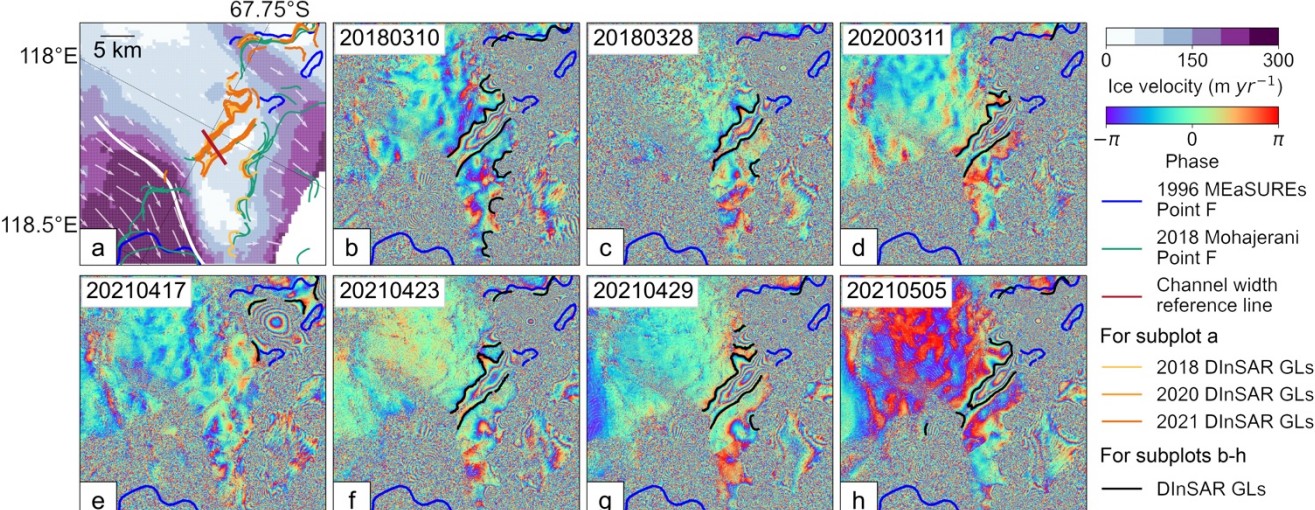

**Figure 8. a) Distribution of different grounding line (GL) products at Moscow University Western Ice Shelf (MUWIS) ocean channel overlaid with the ice surface velocity magnitudes (Rignot et al., 2017) and ice flow directions (white arrows). The reference line for measuring channel width is shown as the red solid line. The Sentinel-1a/b DInSAR GLs mapped in this study are color-coded by year. b-h) GLs (black solid lines) delineated from the Sentinel-1a/b DInSAR interferograms between 2018 and 2021 in this study. In all subplots, the 1996 DInSAR-derived GLs (Rignot et al., 2011, 2016) are shown as blue solid lines. The ice flowline used to measure the GL migration rates is shown as the white solid line and the 2018 DInSAR-derived GLs (Mohajerani et al., 2021) are shown as green solid lines in subplot a).**

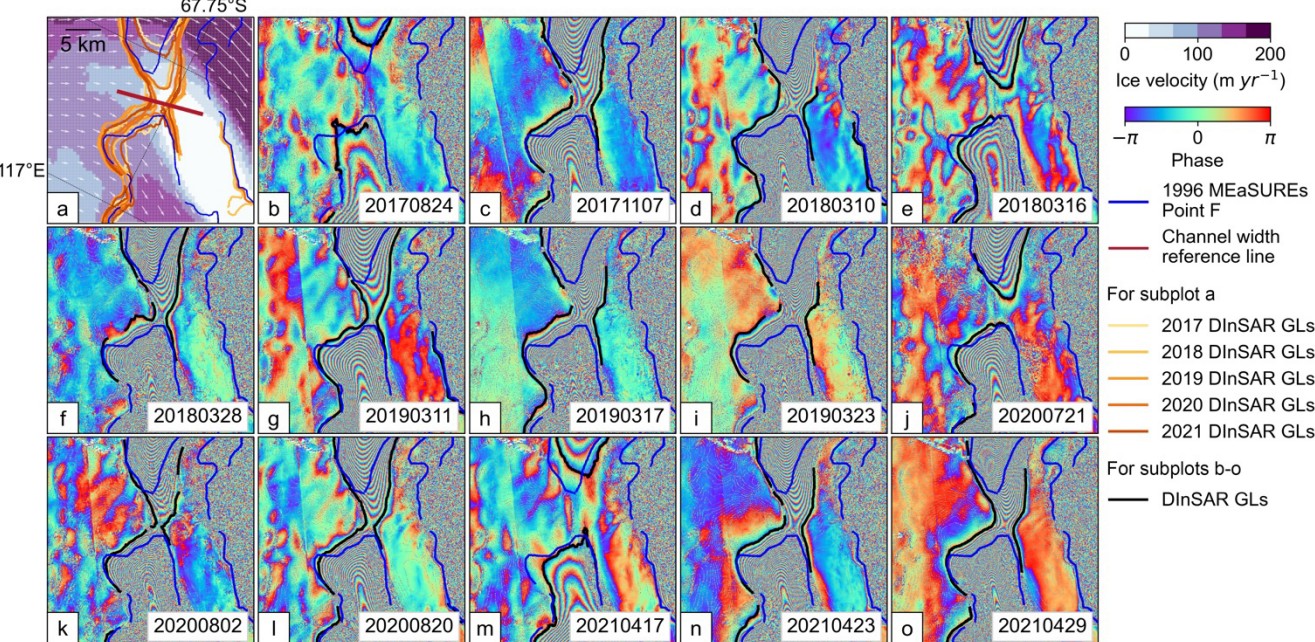

**Figure 9. a) Distribution of different grounding line (GL) products at Totten Glacier Eastern Ice Shelf (TGEIS) ocean channel overlaid with the ice surface velocity magnitudes** (Rignot et al., 2017) **and ice flow directions (white arrows). The reference line for measuring channel width is shown as the red solid line. The Sentinel-1a/b DInSAR GLs mapped in this study are color-coded by year. b-o) GLs (black solid line) delineated from the Sentinel-1a/b DInSAR interferograms between 2017 and 2021 in this study. In all subplots, the 1996 MEaSUREs DInSAR-derived GLs (Rignot et al., 2011, 2016) are shown as blue solid lines.**

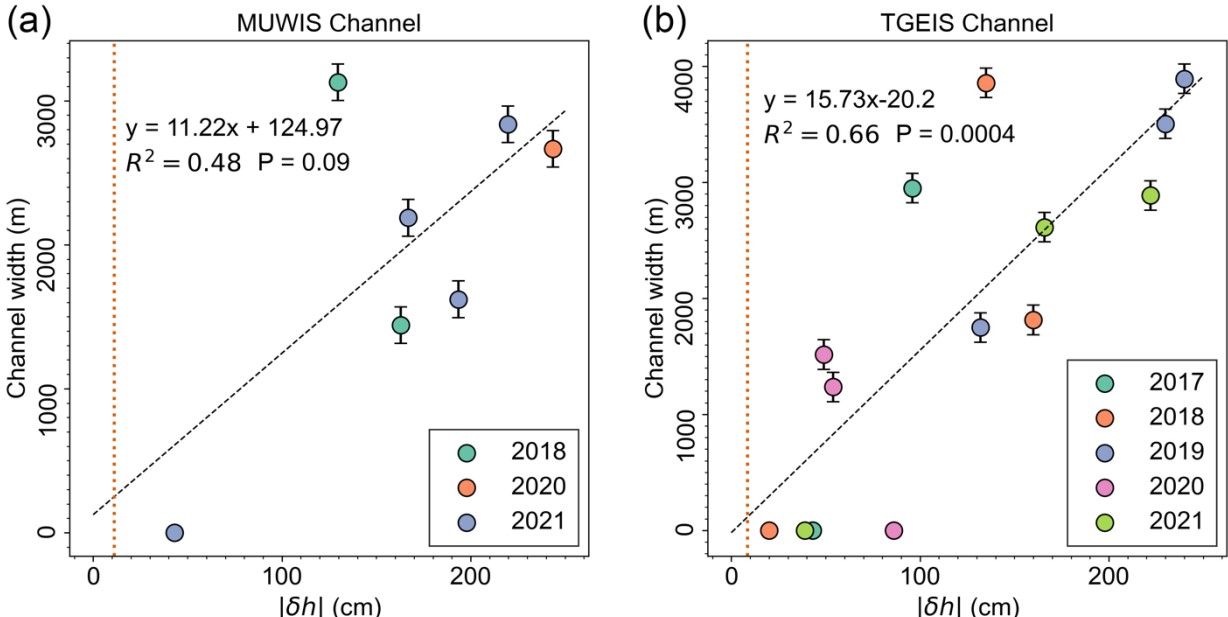

Figure 10. a) Comparison between the absolute differential tidal predictions $|\delta h|$ from the CATS2008 tidal model and the width of Moscow University Western Ice Shelf (MUWIS) ocean channel (Table S6). Year acquisition of DInSAR data is colour-coded, the channel width is measured using the reference line shown as the red line in Figure 8a. b) Comparison between the absolute differential tidal predictions $|\delta h|$ using the CATS2008 tidal model and the width of Totten Glacier Eastern Ice Shelf (TGEIS) ocean channel (Table S7). Year acquisition of DInSAR data is colour-coded, the channel width is measured using the reference line shown as the red line in Figure 9a. The locations of tidal prediction reference points at both regions are shown in Figure S7. In both subplots, the vertical orange dotted lines denote the absolute differential tidal predictions from the CATS2008 tidal model at the time of the satellite image passes used in the MEaSUREs 1996 ERS-1/2 DInSAR GL measurements (Rignot et al., 2011, 2016) (Table S2).

### 3.3 Hydrostatic potential and ice thinning

The GL migrations at the five studied regions across Wilkes Land (Table 1) are happening in areas with high hydrostatic potential, where the ice is only grounded a few tens of meters above hydrostatic equilibrium (Fig. 11). These areas are prone to rapid GL retreat with only a small amount of ice thinning (Li et al., 2015) and large short-term GL fluctuations due to ocean tide variations (Milillo et al., 2017; Brunt et al., 2011). Four distinct regions among those studied, including the MUIS, TGIS, TGEB and MUWIS, are in fast-flowing regions with an ice velocity higher than 200 m yr$^{-1}$. In addition, CryoSat-2 surface elevation change rates during 2010-2019 (Fig. 12) show strong thinning signals at the fast-flowing regions upstream

of the GZs at MUIS, TGIS and TGEB. The highest elevation change rates on the central trunk of MUIS, TGIS and TGEB

are -0.9 ± 0.01 m yr$^{-1}$, -1.64 ± 0.01 m yr$^{-1}$ and -0.72 ± 0.02 m yr$^{-1}$, respectively. The elevation change due to firn and SMB processes is less than 0.16 m yr$^{-1}$ between 2010 and 2016 based on the RACMO 2.3 Firn Densification Model (FDM) (Melchior Van Wessem et al., 2018; Kuipers Munneke et al., 2015). Therefore, it is likely that the surface elevation change is predominantly caused by dynamical thinning. The bed topography underneath MUIS and TGIS main glacier trunks near the GL is 2000 – 2300 m below sea level (Figs. S9a-b). For TGIS, the bed topography remains nearly level along the

southern lobe (Fig. S9b). For the western flank of the MUIS main glacier trunk (Fig. S9a), the TGEB (Fig. S9c) and the MUWIS (Fig. S9d), the bed elevation decreases further inland, and the GLs have been retreating along retrograde bed slopes.

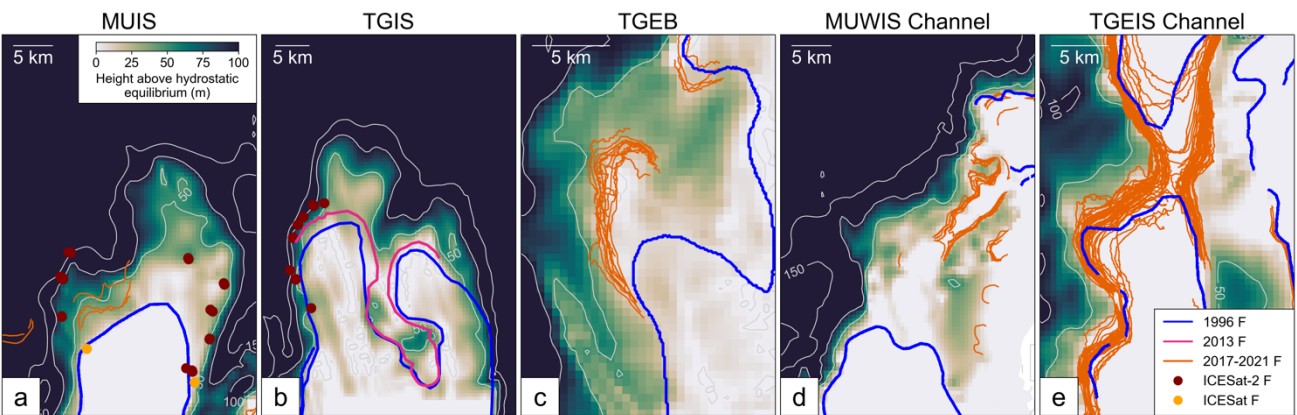

**Figure 11. Height above hydrostatic equilibrium in Moscow University Ice Shelf (MUIS) main glacier trunk (a), Totten Glacier Ice Shelf (TGIS) main glacier trunk (b), Totten Glacier East Branch (TGEB) (c), Moscow University Western Ice Shelf (MUWIS)**

**ocean channel (d), and Totten Glacier Eastern Ice Shelf (TGEIS) ocean channel (e). The DInSAR GLs between 2017 and 2021 mapped in this study are shown as the orange solid lines. The 1996 and 2013 MEaSUREs GLs are shown as blue and pink solid lines, respectively (Rignot et al., 2011, 2016). The ICESat-2-derived Point F locations and ICESat-derived Point F locations derived in this study are shown as maroon and orange dots, respectively.**

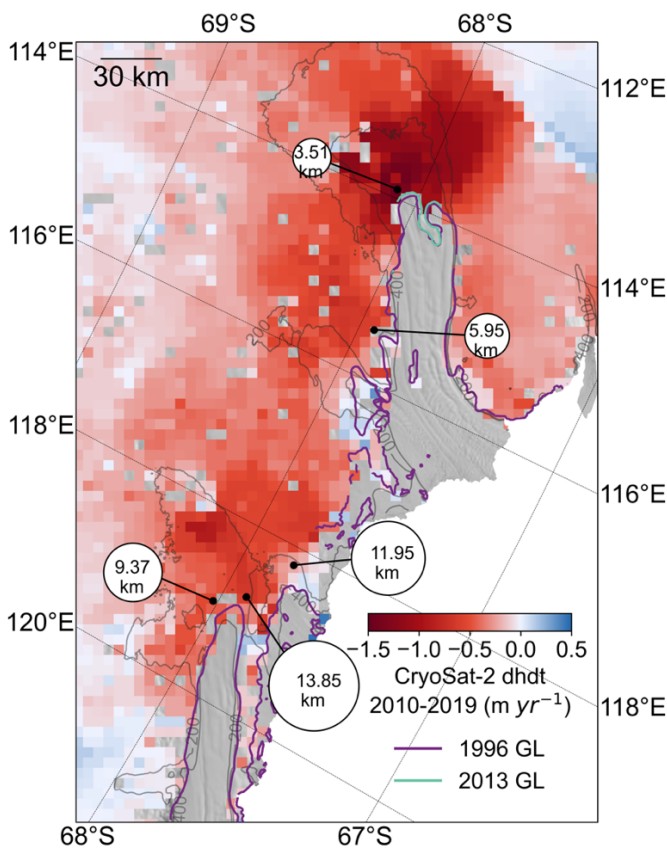

**Figure 12. CryoSat-2 surface elevation change rate between 2010 and 2019 on grounded ice overlaid with the mean grounding line (GL) retreat distances since 1996 measured in this study marked as the white circles (Table 1). The size of the white circle indicates the GL retreat magnitude.**

**Table 1. The mean grounding line (GL) migration distance and standard deviation in five studied regions including Moscow University Ice Shelf (MUIS) eastern flank, MUIS western flank, Totten Glacier Ice Shelf (TGIS) southern lobe, Totten Glacier East Branch (TGEB), and Moscow University Western Ice Shelf (MUWIS).**

| Region | Time Period | Instrument | GL Feature Type | Number of GL Measurements | Mean GL Migration (km) | Standard Deviation (km) |
|---|---|---|---|---|---|---|
| MUIS eastern flank | 1996-2020 | ICESat-2 | Point | 8 | 9.37 | 1.04 |
| MUIS western flank | 1996-2020 | ICESat-2 | Point | 2 | 13.85 | 0.08 |
| TGIS southern lobe | 1996-2020 | ICESat-2 | Point | 12 | 3.51 | 0.49 |
| TGEB | 1996-2020 | Sentinel-1a/b | Line segment | 11 | 5.95 | 0.75 |
| MUWIS | 1996-2021 | Sentinel-1a/b | Line segment | 1 | 11.95* | - |

**\*Note the GL migration distance for the MUWIS is not a mean value as there is only one present-day Sentinel-1a/b DInSAR interferogram available.**

## 4 Discussion

### 4.1 Grounding line retreat and ice dynamics

Although we are unable to duplicate the DInSAR interferogram at the main glacier trunk of MUIS in 1996, the ICESat repeat track analysis along RGT 323 in this study (Fig. S1) shows that the ICESat-derived Point F locations in 2004-2005 are close to the MEaSUREs GL in 1996, which gives us confidence that the 1996 GL is correct. Therefore, we calculated the GL retreat rates by comparing the GLs mapped in this study with the 1996 MEaSUREs GL. The GL retreat rates based on ICESat-2-derived Point F locations and the 1996 MEaSUREs GL are 0.36 – 0.46 km yr$^{-1}$ along the ice flowlines at the

eastern flank of MUIS (Table S3). This contradicts the highest retreat rate of 0.079 ± 0.029 km yr$^{-1}$ in this region calculated by Konrad et al. (2018), which derived the GL migration rate based on a hydrostatic equilibrium assumption using CryoSat-2-derived surface elevation and the Bedmap-2 bed elevation at the MEaSUREs GL locations (Rignot et al., 2011), instead of mapping the GL location directly. However, on the western flank of MUIS, the GL retreat rate from ICESat-2 between 1996 and 2020 is around 0.57 km yr$^{-1}$, at a similar magnitude to the highest retreat rate of 0.25 ± 0.099 km yr$^{-1}$ observed by

Konrad et al. (2018). The bed elevation in MUIS GZ has large uncertainty especially on the eastern flank of MUIS due to the lack of airborne radar data coverage (Fig. S32 in Morlighem et al. (2020)), where the difference between BedMachine and Bedmap-2 bed elevations can be up to 1 km. Therefore, the approach used in Konrad et al. (2018) is likely to have large uncertainties in regions like MUIS, which might cause discrepancies with our GL retreat rates. For Track 841 GT1L (Fig. 4d), the modelled tidal phases are the opposite of the ICESat-2-derived elevation anomalies, although the identified Point F

locations are close to the Sentinel-1a/b DInSAR-derived GL in 2018 by Mohajerani et al. (2021) (Fig. 2a). The reason for this tidal phase difference is possibly because the ice at the deep GZ cannot respond adequately in phase with ocean tides (Reeh et al., 2000) at narrow ice shelf embayment such as MUIS. The ice thickness at the MUIS western flank GZ is about 2500 m (Morlighem et al., 2020; Morlighem, 2020) while the glacier ice thickness at the GL in Reeh et al. (2000) is only about 700 m.

Zero-mean annual ice velocity time series of six different GZ sites between 2005 and 2018 (red box in the grey inset of each subplot) are shown in Figure 13, they were calculated as the deviation from mean ice velocity. In addition, the annual cumulative iceberg calved areas derived from multisource optical and SAR images are also shown (Qi et al., 2021). Both the western and eastern flanks of the MUIS GZ showed ice velocity acceleration by about 50 m yr$^{-1}$ during 2007-2010 and remained almost constant until 2016 (Figs. 13a-b). The iceberg calving at the ice shelf front, and the melt-driven thinning

concentrated near the GL and the reduction of basal traction due to GL retreat, can lead to the speedup of grounded ice (Rignot, 2006; Fürst et al., 2016; Gudmundsson et al., 2019; Joughin et al., 2021). This ice velocity acceleration was unlikely to be caused by iceberg calving, because the sizes of individual iceberg calving events between 2005-2009 are all small-to-medium with an area below 25 km$^2$ and the annual cumulative calving areas during 2007-2010 are less than 50 km$^2$ (grey boxes in Figures 13a-b) (Qi et al., 2021). These small calving events should locate inside the "safety band" of MUIS (Fürst

et al., 2016; Reese et al., 2018), the region where mass loss will not cause major dynamic changes at the GZ. Therefore, a

possible explanation for this discrepancy between ice velocity acceleration and small calving events might be melt-driven GL retreat. Rignot et al. (2019) shows that MUIS lost an integrated mass of 93 Gt during the past four decades, equivalent to a sea-level rise of 0.3 mm. The ice surface elevation at the fast-flowing region has also been decreasing significantly during the past two decades (Smith et al., 2020), similar to the negative surface elevation change between 2010-2019 observed in our study (Fig. 12). The western flank of the MUIS GZ is located at a retrograde bed slope (Fig. S9a), therefore current ice thinning trends might contribute to a continuous GL retreat in future.

The southern lobe of TGIS retreated at a mean rate of 0.21 km yr$^{-1}$ in 2013-2020 (Table S4), higher than the 0.12 km yr$^{-1}$ mean retreat rate in 1996-2013 mapped by Li et al. (2015). The ice velocity at the southern lobe of TGIS first increased from 2005 to 2007 by 51 m yr$^{-1}$, kept constant with a slight decrease of 9 m yr$^{-1}$ until 2016, then decreased 102 m yr$^{-1}$ in 2017 (Fig.13c). Despite this recent slowdown, the mass loss of TGIS has increased through the period of 1979-2017 and remained at about 10% of the balance flux (Li et al., 2016; Rignot et al., 2019). This would indicate a continuous GL retreat at the fast-flowing region of TGIS. Note that we only managed to map the GZ at the eastern flank of the TGIS southern lobe where the bed elevation is increasing inland along the ice flowlines (Fig. S2b and S9b), this might not represent the overall GL retreat rates over this region despite the highest negative elevation rates derived from CryoSat-2 radar altimetry (Fig. 12)

Sentinel-1a/b DInSAR interferograms show that GL retreat also happened at TGEB with a mean rate of 0.25 km yr$^{-1}$ from 1996-2020 (Fig. 7), similar to the modelled results on Totten Glacier's evolution through 2100 (Pelle et al., 2021). Our average retreat rate is almost ten times the highest GL retreat rate of 0.027 ± 0.016 km yr$^{-1}$ during 2010-2016 calculated by Konrad et al. (2018) in this region. The GL retreat at TGEB happened along a retrograde bed slope (Fig. S9c), and coincided with high ice velocity (Fig. 6a) and negative elevation change in this region (Fig. 12). Moreover, the ice velocity at the GZ has been increasing moderately since 2007 by about 40 m yr$^{-1}$ (Fig. 13d). This behavior suggests the possibility of MISI taking place in this sector and that the GL retreat possibly occurred during the past decade.

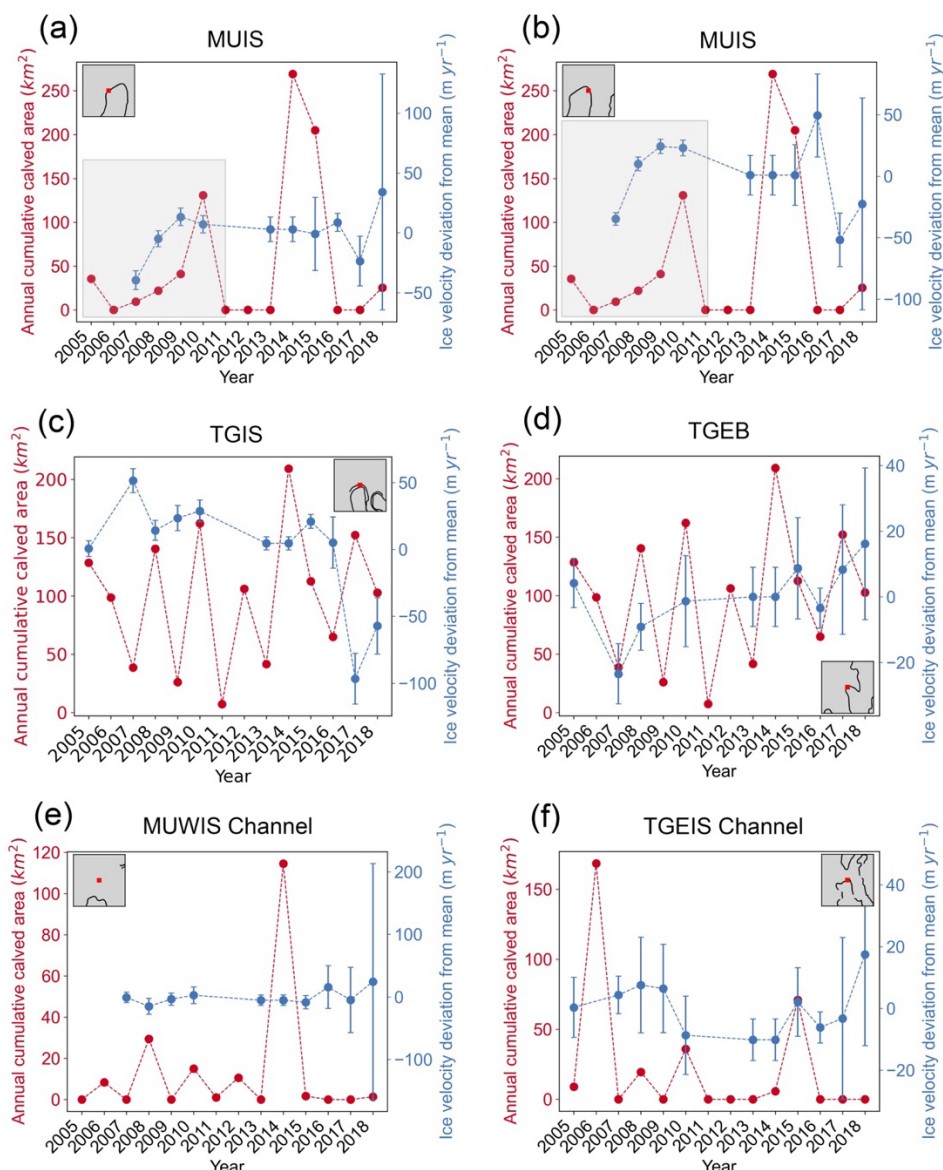

**Figure 13. The zero mean ice velocity time series at each grounding zone (red box in the grey inset map) derived from MEaSUREs (2005-2016)** (Mouginot et al., 2017a, b) **and ITS_LIVE (2017-2018)** (Gardner et al., 2018, 2019) **annual ice velocities, and the annual cumulative iceberg calved area at the ice shelf front (Qi et al., 2021).**

## 4.2 Ocean channels and ocean-induced basal melting

The large GL retreat of 11.95 ± 0.13 km from 1996-2021 at MUWIS is located at an ice plain with a low height above flotation (Fig.11d), a high ice velocity (Fig. 8a) and small surface elevation change (Fig.12). In comparison, the observed GL

migration here conflicts with the GL advance predicted in Konrad et al. (2018). Our results also reveal the existence of a tide-modulated ocean channel with a water depth deeper than 600 m below sea level (Figs. 8, 14 and S9d). Warm mCDW has been observed at the Sabrina Coast continental shelf in front of TGIS (Fig. 1), and can reach the TGEIS cavity through a deep trough to drive rapid basal melt (Fig. 9) (Rintoul et al., 2016; Greenbaum et al., 2015; Silvano et al., 2017). This poleward transport is possibly caused by a combination of wind forcing, decreased polynya activity and cyclonic eddies

(Greene et al., 2017; Gwyther et al., 2014; Hirano et al., 2021). Similarly, the ocean heat transfer by mCDW to MUIS depends on the nearby Dalton polynya activity and the flow route of the Antarctic Slope Current (ASC) across the Dalton Rise (Gwyther et al., 2014). Low polynya activity and an overflow over the Dalton Rise will result in a strong inflow of warm water into the MUIS cavity. The newly discovered MUWIS ocean channel, which connects MUWIS and TGEIS, provides a route for mCDW to reach MUWIS from the neighboring TGEIS cavity (green arrow in Figure 14). Similar to the

mechanism proposed by Greenbaum et al. (2015) on the retreat of the TGEIS GZ, the GL retreat at MUWIS is therefore possibly caused by the increased ice shelf basal melting due to ocean heat transferred through this ocean channel, especially in a region of high hydrostatic potential (Fig. 11d). If thinning continues, the GL will retreat to a retrograde bed slope just south of the current GL indicated by the red arrow in Figure 14, which has the potential to trigger a regional MISI. This will result in further GL migration and contribute to the destabilization of the MUIS sector in the future.

While Sentinel-1a/b DInSAR interferograms show the existence of an ocean channel at TGEIS similar to the findings of Greenbaum et al. (2015), the channel width is highly correlated with ocean tide variations (Fig. 10a). In contrast to the prediction of a long-term channel widening caused by ice thinning in their study, the largest channel width observed in this study is 3.89 ± 0.13 km (Table S7), comparable to the ~4 km channel width measured in 2008-2012 by Greenbaum et al. (2015). In addition, Figure 9 shows that the TGEIS ocean channel closes at a low differential tidal amplitude. If this short-

term opening and closing was the case in 1996, it is likely that the 1996 MEaSUREs GL was accurate since the reconstructed differential tidal amplitudes from both the CATS2008 (orange vertical dotted line in Figure 10b) and FES2014 tidal models (Table S3) were low and the channel could be closed as a result.

As the weak underbelly of East Antarctica, Wilkes Land has been losing mass at accelerated rates due to warm water intrusion (Rignot et al., 2019; Pelle et al., 2020). Our results show not only that the GL at the southern lobe of TGIS keeps

retreating at a comparable rate to 1996-2013, other regions, including TGEB, MUIS and MUWIS, show pervasive GL retreats along the retrograde bed slopes at the fast-flowing regions. Moreover, an ocean channel at MUWIS discovered in this study can provide a pathway for ocean heat into the MUIS main ice shelf cavity. Although we are unable to identify the exact time stamps when the GL started retreating during the past two decades for MUIS, multiple lines of evidence show that MUIS and MUWIS are susceptible to potential MISI and are currently contributing to sea-level rise.

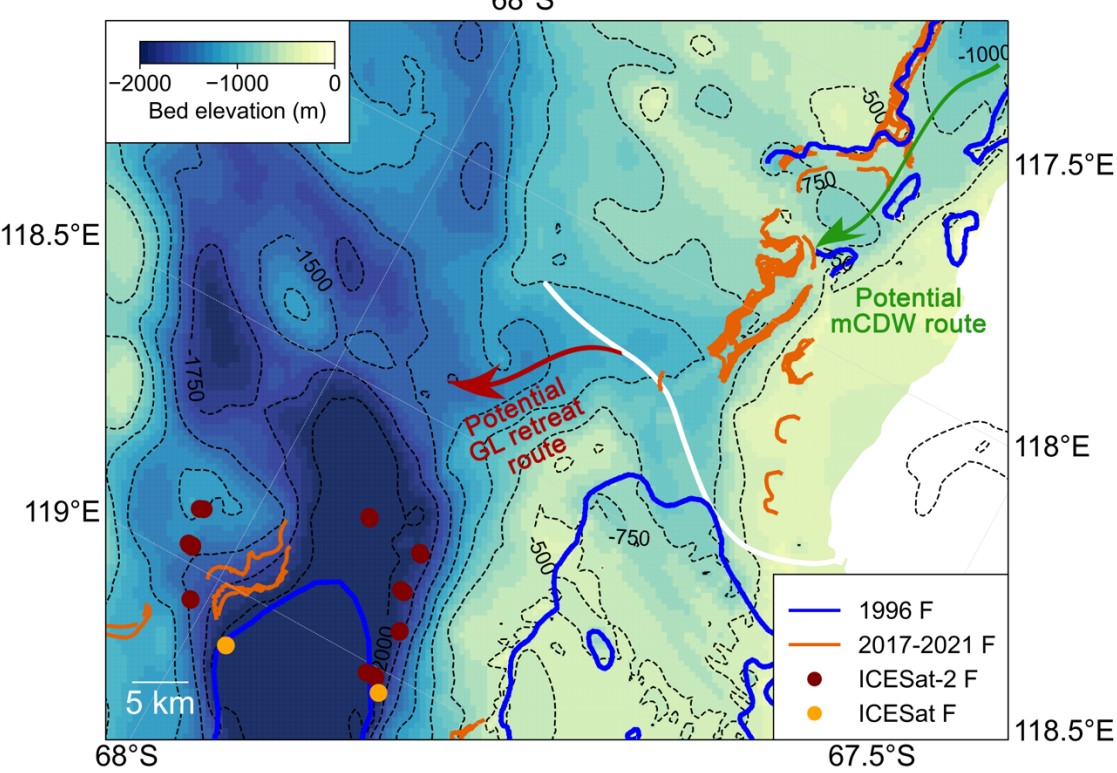

**Figure 14. The potential grounding line retreat route at the Moscow University Western Ice Shelf (MUWIS) along the retrograde bed slope is marked by the red arrow into the main glacier trunk of Moscow University Ice Shelf (MUIS). The potential warm modified Circumpolar Deep Water (mCDW) route from Totten Glacier Eastern Ice Shelf (TGEIS) is marked by the green arrow. All map features are overlaid on the BedMachine bed elevation (Morlighem et al., 2020).**

## 5 Conclusion

By combining Sentinel-1a/b DInSAR interferograms and ICESat-2 laser altimetry data, we found pervasive GL retreats between 1996 and 2020 along the ice plains at the central glacier trunks of Totten and Moscow University glaciers, which drain a large portion of the marine-based Aurora Subglacial Basin in East Antarctica. The mean GL retreat distances in 1996-2020 at the eastern and western flanks of Moscow University Glacier's main glacier trunk are 9.37 ± 1.04 km and 13.85 ± 0.08 km, respectively. Meanwhile during the same period, the mean GL retreat distances are 5.95 ± 0.75 km and 3.51 ± 0.49 km at TGEB and the southern lobe of the TGIS main glacier trunk, respectively. The GL retreats are concentrated along fast-flowing regions with observed high thinning rates over the past 10 years, indicating a dynamical mass imbalance, which is likely related to the presence of warm mCDW in this region. GL retreats at MUIS and TGEB are along the retrograde bed slopes, accompanied by ice velocity acceleration and negative surface elevation change, indicating that a regional MISI is likely underway. The analysis of Sentinel-1a/b DInSAR interferogram time series revealed a tide-

modulated ocean channel located at MUWIS. The opening of this channel connects the two separate MUIS and TGIS systems, which might open a pathway for warm mCDW to enter the MUWIS cavity from the TGEIS cavity. This could facilitate further GL retreat at MUWIS into a retrograde bed slope and merge into the main glacier trunk of MUIS in the future. Despite these GL migrations, our study finds that the TGEIS ocean channel has not further expanded as suggested by Greenbaum et al. (2015), and its width is positively correlated with the ocean tidal range.

This paper is the first investigation of both short-term and long-term GL changes on Totten and Moscow University Glaciers. Although we are unable to identify the exact time stamps when GLs started retreating, the findings offer new insights into the current GL and ice sheet instability in Wilkes Land, a region that has the potential to make a significant contribution to future sea-level rise. Additionally, this study highlights the importance of using a combination of different Earth observation techniques in monitoring the GL locations, especially in separating short-term tidal variability from the long-term GL migrations. With the advent of more ICESat-2 data in coming years as well as the new NASA-ISRO SAR (NISAR) Mission planned for 2024, routine monitoring of this critical region with improved coverage will be possible.

**Data availability**

The ICESat-2 and ICESat surface elevation data used in this study are available from the National Snow and Ice Data Center (NSIDC). The CryoSat-2 Swath mode thematic point product was processed by CryoTEMPOEOLIS (https://www.cryotempo-eolis.org) and is distributed by the European Space Agency (ESA) (https://science-pds.cryosat.esa.int/). The Sentinel-1a/b data is provided by the ESA and is accessible from the Alaska Satellite Facility (ASF) via the ASF API (https://asf.alaska.edu/api/). The grounding line data produced in this study will be available via a DOI generated at the University of Bristol data repository, data.bris.

**Author contribution**

TL designed the study, produced the results, and wrote the paper. GJD assisted with DInSAR processing. SJC produced the CryoSat-2 elevation change. JLB conceived the study and contributed to the interpretation of the results and paper writing. All authors commented on the manuscript.

**Competing interests**

The authors declare that they have no conflict of interest.

## Acknowledgements

Tian Li received funding from the China Scholarship Council (CSC)–University of Bristol joint-funded PhD scholarship. Jonathan L. Bamber and Stephen J. Chuter received funding from the European Research Council (GlobalMass; grant no. 694188). Stephen J. Chuter also received funding from the European Space Agency (ESA) as part of the Climate Change Initiative (CCI) fellowship (ESA ESRIN/Contract No. 4000133466/20/I/NB). Jonathan L. Bamber also received funding from the German Federal Ministry of Education and Research (BMBF) in the framework of the international future lab AI4EO (grant no. 01DD20001). We thank Michiel van den Broeke and Peter Kuipers Munneke for providing the RACMO 2.3 data. We thank Chad A. Greene, Bernd Scheuchl, Bert Wouters, Rob Bingham, Pietro Milillo and one anonymous reviewer for providing valuable comments on this study.

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
