# Peer review of "Grounding line retreat and tide-modulated ocean channels at Moscow University and Totten Glacier ice shelves, East Antarctica"

_The Cryosphere, 2022_

## Author Response (AR1)

**Author response by Tian Li on behalf of all the other authors:**

We would like to thank the reviewers for their detailed and constructive comments on our paper, which have proven very helpful in revising this manuscript. We have been able to fully address the suggested changes and we think the revised manuscript has been greatly improved. Our detailed responses to each comment presented by the reviewers are in blue text.

**Reviewer 01**

The Authors use Sentinel-1a/b Interferometry combines with ICESat-2 laser altimetry between 1996 and 2020 to monitor grounding line retreat over coherent parts of the Moscow University and Totten Glaciers. They combine bed topography, ice velocity maps, tidal information and historical grounding line position to map both long term grounding line retreats and tidally induced grounding line retreats. As stated in the conclusions, I would not consider this paper a "*comprehensive investigation of both short-term and long-term GL changes*", since the InSAR data are decorrelated over key areas (main trunk) over these two glaciers, but more a confirmation of the tidally induced grounding line migration dynamics.

Nevertheless, the paper is well written, fits the topic of the journal and I would like to see it published on The Cryosphere.

Thank you for the constructive comments on the manuscript, we have now removed "comprehensive" from the Conclusion section.

I have some recommendations before accepting this paper for publication:

1. The authors states that "*large uncertainties might exist for the MEaSUREs GL in 1996 due to tidal amplitude variations*". I believe this sentence is not accurate because this is not about the measurement accuracy but measurement availability as a function of the tidal cycle. A single InSAR GL measurement can have accuracies between 100-200 meters. Hence, I would rephrase this with something like "*The 1996 GL MEaSUREs dataset only provides one grounding line measurement which does not allow us to characterize the tidally induced grounding line variability*".

Thank you for the suggestion. Agree and we have now changed this sentence to "It is important to note that the MEaSUREs GL dataset in 1996 only provides one GL measurement in areas not covered by the ICESat repeat tracks, which does not allow characterizing the tidally induced GL variability."

2. For some reasons the authors did not include these dataset in the tidal plots (Figure 3,7,9) The authors state that they did not reprocess the ERS dataset because of the "*lack of available Level1 ERS-1/2 SAR SLC data in this region*". Perhaps you could look at the Level0 raw data and check in the annotation files the acquisition time? This would enable to put in context the 1996 grounding line as a function of tidal levels.

Thank you for the advice. We have now checked the acquisition times of the 1996 MEaSUREs GL measurements from the ESA ERS-1/2 online catalogue based on the satellite orbit and date information available from the MEaSUREs GL data product. We calculated the differential tidal amplitudes from both CATS2008 and FES2014 tidal models for all the five studied regions in this study, please see Table S2.

We also labelled the modelled differential tidal amplitudes for the 1996 GLs from the CATS2008 tidal model for Moscow University Ice Shelf (MUIS) in Figure 3 and Totten Glacier East Branch (TGEB) in

Figure 7 to show the tidal level of the 1996 grounding line. For Totten Glacier Ice Shelf (TGIS) in Figure 5b, we calculated and labelled the modelled tidal range for the four ERS-1/2 satellite image passes in 1996 from the CATS2008 tidal model according to Table S2. The reason of using tidal range here instead of the differential tidal amplitudes is that Figure 5b shows the ICESat-2-derived tidal range – the difference between the maximum and minimum elevation anomalies of ICESat-2 repeat tracks.

3. Lines 200-203: The authors find a high positive correlation with the absolute tidal range, I think it would be worth highlighting this goes against (Tsai and Gudmundsson 2015) theory indicating an asymmetric migration between high and low tides. Moreover, It is not clear to me why (as an example) to a negative low tide of 1 meter should correspond an inland retreat of the grounding line equal to a positive 1m high tide.

Thank you for the comment.

Tsai and Gudmundsson (2015) used an elastic fracture model to simulate the tidal grounding line migration over a tidal cycle and showed that grounding line migration is inherently asymmetric and nonlinear – the GL migration upstream is about ten times further as tide rises than its downstream migration as tide falls. This asymmetry is partly controlled by the nonlinear relationship between pressure forcing and fracture growth. Although their model does not rely on hydrostatic equilibrium assumption on the groundline line, their experiments show that upstream GL migration at high tide is still larger than the downstream GL migration at low tide.

The linear relationship between the differential tidal amplitude and the GL migration since 1996 in Figure 3, however, shows a near symmetric GL migration with ocean tides and contradicts the conclusion in Tsai and Gudmundsson (2015). One possible reason that could contribute to this contradiction is that we only have three DInSAR GL migration samples over the MUIS GZ, which means the linear relationship is not significant. More DInSAR GL mappings at MUIS are needed to draw a robust conclusion about this symmetric GL migration with ocean tides. In addition, the bedrock slope is poorly known at MUIS GZ (Morlighem, 2020; Morlighem et al., 2020) which may influence the model output.

We highlighted this contraction in Line 234 – 237 : "This symmetric GL migration over a 12-day period, however, contradicts the nonlinear and asymmetric GL migration over a tidal cycle simulated by Tsai and Gudmundsson (2015) using an elastic fracture model. The contradiction can be possibly attributed to the fact that we only have limited three DInSAR GL measurements over this region and the bedrock elevation at MUIS GZ is poorly known (Morlighem, 2020; Morlighem et al., 2020) which may influence the model output from Tsai and Gudmundsson (2015)."

4. Lines 222-223 The authors find evidence of tidally modulated subglacial lakes, but do not show the bed potential that would accommodate the creation of these lakes.

Thank you for the advice. We have now added the Figure S6 in the supplementary material to show the height above hydrostatic equilibrium of these two subglacial lakes at the Moscow University Ice Shelf.

[Figure]

Figure S6. Height above hydrostatic equilibrium at the two subglacial lakes near Moscow University Ice Shelf (MUIS) main glacier trunk.

5. Lines 259-260 Authors mention Points F and H but I was not able to locate them in the figures. Perhaps it would make sense to add them also in the Primary figures and not in the supplementary material.

Thank you for the advice. We wanted to make sure the focus of the figures in the main manuscript is the grounding line (Point F) locations and their changes through time. The Point H is mainly used to calculate the tidal ranges at ICESat-2-derived GZs for MUIS and TGIS, therefore we would prefer not to include the distributions of Point H in the main text.

6. The authors find no correlation between tides and GL migration over Totten East. I do not have enough elements to assess this result since no map showing where the CATS 2008 reference point has been taken is present in the manuscript. If the authors chose the same reference point for both glaciers then this could explain why one glacier is uncorrelated. The authors should display these info, describe them in the text and justify the reasons behind their choice.

Thank you for the advice. We have now added the Figure S7 in the supplementary material to show the distributions of the reference points used when calculating the tidal amplitudes from the CATS2008 and FES2014 tidal models. The reason of choosing the tidal reference points for FES2014 off coast is this tidal model does not provide tidal predictions on the ice shelf areas.

[Figure]

Figure S7. The spatial distributions of the CATS2008 reference points (circles) and FES2014 references points (triangles) used in the DInSAR tidal amplitude predictions for each studied region. Note FES2014 tidal model does not provide tidal predictions over the ice shelves, its tidal prediction reference points are located off coast.

Tsai, V. C., & Gudmundsson, G. H. (2015). An improved model for tidally modulated grounding-line migration. Journal of Glaciology, 61(226), 216-222.

Pietro Milillo

**Reviewer 02**

This study uses a suite of remote sensing datasets and techniques to delineate the grounding line, or a grounding line proxy, at Totten and Moscow University glaciers, and to calculate change in grounding line position over time. These datasets include Sentinel1 a/b SAR interferograms, and ICESat and ICESat-2 along-track surface elevations. They find that grounding lines have retreated by 3-13 km at varying rates among the different study areas. They describe a newly identified tidally-modulated ocean pathway between Totten ice shelf and Moscow University ice shelf that could contribute to future mCDW interaction with deep grounding lines and continued retreat down retrograde bed slopes.

Overall, I find the study impactful and rigorous, and worthy of inclusion in this journal, although I have some moderate concerns. I think they can be addressed by some restructuring within the methods, results, and discussion sections.

We would like to thank the reviewer for the detailed and constructive comments which have greatly improved this paper.

In the Methods section, it becomes difficult to follow which GL/GZ positions were generated in this study vs which were used for comparison, although historical GLs are correctly attributed. There is no mention in Section 2.1 of the attempt to replicate the 1996 ERS-1/2 GL, nor in Section 2.2 of the comparison between the 1996 GL and the ICESat-derived GZ discussed at the beginning of Section 3.1. (Was replication of the 1996 GL also attempted for TGIS?) Furthermore, the 1996 and the Mohajerani GL positions are repeatedly compared to ICESat-2-derived GZ positions, although the methodology of this comparison is not present in Section 2. This could be improved by including a description of how the flowlines were selected and how calculations of retreat rates between and among different datasets are unique, perhaps in a new section entitled "Grounding line/zone migration rates".

Thank you for the advice.

In Section 3.1.2, we mentioned that "The historic GLs in 1996 and 2013 in this region were mapped with high confidence according to Figure 2 in Li et al. (2015), therefore they were used as a reference in calculating GL migration rates with ICESat-2-derived Point F locations along the ice flowlines shown in Figure S2b". Therefore, the replication of 1996 GL is not attempted for TGIS.

In Section 2.2 "Grounding zone mapping from ICESat", we added "The newly-mapped ICESat-derived Point F locations at MUIS were also used to validate the historic DInSAR-derived GLs from the MEaSUREs project (Rignot et al., 2011, 2016)" to clarify the purpose of mapping ICESat Point F at MUIS.

We added a new Section 2.4 "Grounding line migration distances" to talk about the selection of ice flowlines and the calculation of grounding line migration distances:

*2.4 Grounding line migration distances*

*The GL migration distances were measured at MUIS, TGIS, TGEB and MUWIS along ice flowlines by comparing the present-day GLs mapped in our study with the 1996 ERS-1/2 DInSAR-derived GL from the MEaSUREs project (Rignot et al., 2011, 2016). For MUIS and TGIS, the GL migration distances were calculated as the along-flowline separations between the selected ICESat-2-derived Point F locations and the 1996 MEaSUREs GL (Fig. S2). At these two regions, the ice flowlines were selected based on two criteria: 1) passing through the ICESat-2-derived Point F locations, and 2) having intersections with the 1996 MEaSUREs GL. In addition, the selected ice flowlines should be near parallel to each other and close in space to reduce uncertainties in GL migration rates due to discrepancies in spatial distribution. At MUIS, the GL migration distances were also measured by comparing the Sentinel-1a/b*

*DInSAR-derived GLs with the 1996 MEaSUREs GL along an ice flowline, which should locate at the glacier central trunk and has intersections with both Sentinel-1a/b DInSAR GLs and the MEaSUREs 1996 GL. The GL migration distance measurements at TGEB and MUWIS follow the same approach by comparing the Sentinel-1a/b DInSAR-derived GLs with the 1996 MEaSUREs GL.*

I feel that the analysis of the annual cumulative iceberg calved area vs velocity and GL retreat would work best if confined to the discussion section, as the authors are using existing datasets to contextualize the new GL/GZ maps. Specifically, Section 3.3 reads to me entirely as (important!) discussion material rather than novel results. Furthermore, I think readability could be improved by creating subsections within the discussion similar to the sections in the results.

Thank you for the advice. We have now merged the analysis of the annual cumulative iceberg calved area vs ice velocity changes at the GZs into the Discussion Section 4.1 "Grounding line retreat and ice dynamics". We divided the Discussion sections into two subsections:

- 4.1 Grounding line retreat and ice dynamics
- 4.2 Ocean channels and ocean-induced basal melting

Overall, I think the main text figures are chosen and developed appropriately and that they contribute meaningfully to the results and conclusions. In general, I think the figures throughout could be improved by ensuring that consistent colors/symbols and/or axis labels are used for the same datasets or types of data (e.g. historical GLs, flowlines, color scales for dates, etc.). I have highlighted some discrepancies in the minor comments below, but there may be more.

Thank you for the advice. We have now modified the Figures 2, 5, 6, 8, 9, 11, 14, S2, S9 (these are the figure numbers in the revised manuscript) regarding the colours of different types of data, including historic GLs and flowlines, according to the following standard:

| Feature | Symbol |
|---|---|
| 1996 MEaSUREs GLs | blue line |
| 2013 MEaSUREs GLs | pink line |
| 2018 Mohajerani GLs | green line |
| Sentinel-1a/b DInSAR GLs generated in this study | Oranges colormap by image acquisition year when the GLs are overlaid on ice velocity map; solid black lines when the GLs are overlaid on the DInSAR interferograms |
| ICESat-2-derived Point F | maroon circle |
| ICESat-derived Point F | orange circle |
| Ice flowline | white line |
| Ocean channel width reference line | red lines |

Note for better visualization of the GL mapping from Sentinel-1a/b DInSAR interferograms generated in this study, when the background image is the Sentinel-1a/b DInSAR interferogram, we kept the DInSAR GLs delineated in this study as thick black lines because they can immediately stand out from the rainbow colormap of the interferogram.

In addition, we didn't change the colors of historic MEaSUREs GLs in Figures 1 (study region) and 12 (CryoSat-2 elevation change) because the main focuses of these two figures are not grounding line but oceanic thermal forcing for Figure 1 and elevation change rates for Figure 12. And we think the original color choices of the historic GLs in these two figures match better with the background colormaps for either oceanic thermal forcing or CryoSat-2 elevation change rates.

**Specific Comments:**

L 12 - 13 - This sentence indicates retreat from 1996-2020, but above it says that GL locations from 2017-2021 were mapped, and the main text also indicates that GL positions were mapped from the ICESat era. Please revise to include either only the data in the 2017-2021 period, or mention that the 2017-2021 GL positions were compared to historical GL positions.

Agree, we have rephrased this sentence into "We detected pervasive GL retreat along the ice plains at the glacier central trunk of Totten Glacier Ice Shelf (TGIS) and Moscow University Ice Shelf (MUIS), where the GL retreated 3.51 ± 0.49 km and 13.85 ± 0.08 km since 1996, respectively, by comparing the 2017-2021 GL measurements with the historic GLs."

L 91- It is difficult to keep track of which GL positions are generated in this study vs which are historical datasets used for comparison. It might be helpful to include the historical GL position datasets in Section 2.6 to make this abundantly clear.

Agree, we have moved the descriptions on MEaSUREs GL product (Rignot et al., 2011, 2016) and the Mohajerani et al. (2021) GL product to Section 2.6 (now Section 2.7) as additional datasets.

L 127-130 - Is this the description of the methodology in Li et al 2020, 2022? Please make this clear. Furthermore, not all 1387 RGTs have coverage in this region, and the distinction between the six single-beam repeat-track data groups and three beam-pair repeat-track data groups is unclear - could you reference the specific RGT numbers shown in the figures here?

Thank you for the advice. Yes, this is the description of the methodology in Li et al. (2020, 2022). We have changed this sentence into "Ground tracks of four Reference Ground Tracks (RGTs) 163, 841, 1108, 1283 were used at MUIS, while ground tracks of four RGTs 87, 262, 1032, 1207 were used at TGIS. Each RGT has six ground tracks, they were further categorized into nine distinct repeat-track data groups, including six single-beam repeat-track data groups and three beam-pair repeat-track data groups (Figures 4a, b in Li et al. (2020))."

L 125 - 132 - This entire paragraph is difficult to follow. It might be helpful to include more references to explanatory figures, or consider omitting if it is entirely a summary of Li et al., 2020 & 2022

Agree, we have now referenced the explanatory schematics Figures 4a, b and the Equations 1, 2 in Li et al. (2020) when describing the methods regarding repeat-track data groups and the elevation anomaly calculation.

L 177-178  - Please consider including more details about how the tidal ranges are obtained from the tide models, especially since in the results it is mentioned that the tidal ranges are also obtained at GZ positions obtained from ICESat-1/2. Please also consider including a description of the historical GL data here in Section 2.6.

Agree. We added an Eq. (2) to explain the calculation of differential tidal amplitudes $\delta h$ obtained from the tidal models for each DInSAR interferogram:

*"Two different tide models CATS2008 (Padman et al., 2002) and FES2014 (Lyard et al., 2021) were used to calculate the differential tidal amplitudes $\delta h$ at each DInSAR GL measurement using Eq. (2),*

$$\delta h = (h4 - h3) - (h2 - h1),\quad (2)$$

*where h1, h2, h3, h4 are the modelled tidal amplitudes at the acquisition time of each SAR image pass used in the DInSAR interferogram. The tidal range for the DInSAR GL measurement is calculated as the difference between the maximum and minimum tidal amplitudes."*

The **differential tidal amplitude** is different from the **tidal range** used for ICESat and ICESat-2 laser altimetry GL mapping, the latter is calculated as the difference between the maximum tidal amplitude and the minimum tidal amplitude. To make this clear, we used "differential tidal amplitudes δh" for DInSAR GLs and "tidal range" for laser altimetry derived GLs throughout the paper.

In addition, we now included a description of the historical GL datasets in Section 2.6 (now Section 2.7), including the MEaSUREs GL product (Rignot et al., 2011, 2016) and the Mohajerani et al. (2021) GL product.

L 182-185 - it is not clear at this point why you use the cumulative iceberg calved area in your analysis. See general comment above about restructuring to include this information entirely in the discussion section.

Agree and done. We have removed the description of cumulative iceberg calved area from this section and restructured it into the Discussion Section 4.1, please also see our previous response.

L 200-201 - Please specify between which two interferograms the 2.53 km migration is observed, as Table S4 doesn't specify the GL shift between each cycle. To me, it is unclear whether the "GL retreat since 1996 along ice flowline in Figure 2" in Table S4 is the mean or median GL shift since 1996 among the three acquisition dates in each line, or whether each line of Table S4 represents a different interferogram created from three Sentinel 1 scenes (so the interferogram dates in Fig. 2 are nominal based on t2 in Table S4). Please clarify.

The 2.53 km migration is observed between the two interferograms on 23rd and 29th April 2021 (Figs. 2d, e). This sentence is now modified as "Three Sentinel-1a/b DInSAR interferograms with a 6-day repeat cycle (Figs. 2d-f) show a rapid short-term GL migration of 2.53 ± 0.13 km in just six days between 23rd and 29th April 2021 (Table S1)."

Each line of Table S1 (as well as Tables S5, S6, S7 in the revised manuscript) represents a different interferogram created from three Sentinel-1 scenes and the nominal date of each interferogram is t2. We have added the following sentence in the table caption to clarify this:

"Each row represents a different interferogram generated from three different Sentinel-1 image scenes acquired at time stamps of t1, t2 and t3. The nominal time stamp of each interferogram is t2."

L 256-257 - Please specify why both a range in GL retreat and an uncertainty are reported (e.g. from 1996 to different dates in the year 2020?)

We have now removed the uncertainty here.

L 259-260 - "We directly measured the tidal amplitudes…" - Please specify this in the methods section, and please specify how the tidal range in Fig. 5b is derived/is different from the tidal amplitude, and how it differs from $|\delta h|$ in Fig. 3.

Thank you for the advice. The "tidal amplitudes" was wrong here, which should be "tidal ranges". We have modified this sentence into "We directly calculated the tidal range as the difference between the maximum and minimum ICESat-2 elevation anomalies at Point H for each GZ measurement." The

calculation methods of differential tidal amplitude $\delta h$ and the tidal range are now in Section 2.6 (now Section 2.7), please also see our previous reply.

L 275 - It is not clear in Fig. 6 that the GL has been continuously retreating since 1996, or between 2018-2021, and Table S5 indicates that the GL advanced between some cycles. Please revise, and consider using a color scale/different symbols for the DInSAR dates for the GL lines in Fig. 6a.

Agree. We have now distinguished the colours of the Sentinel-1a/b DInSAR GLs mapped in this study by acquisition year in Figure 6a, and modified the colours of the historic GLs to be consistent with other DInSAR plots throughout the manuscript, please also see our previous reply.

[Figure]

Figure 6. a) Distribution of different grounding line (GL) products at Totten Glacier East Branch (TGEB) overlaid with the ice surface velocity magnitudes (Rignot et al., 2017) and ice flow directions (white arrows). b-l) GLs (black solid line) delineated using Sentinel-1a/b DInSAR interferograms between 2018 and 2021. In all subplots, the 1996 MEaSUREs DInSAR-derived GLs (Rignot et al., 2011, 2016) and 2018 DInSAR-derived GLs (Mohajerani et al., 2021) are shown as blue solid lines and green solid lines, respectively. The ice flowline used to measure the GL migration rates is shown as the white solid line in subplot a).

L 301-302 - Please consider including a description of how the channel is identified in the interferograms

The method of identifying the ocean channels from the Sentinel-1a/b DInSAR interferograms is the same with the method of mapping GL from DInSAR interferograms, as the boundary of the ocean channel is also a grounding line. We have modified this sentence into "Two ocean tide modulated channels at the low-lying areas of TGIS and MUIS - one located at MUWIS (Fig. 8, box D in Fig. 1b) and the other located at TGEIS (Fig. 9, box C in Fig. 1b), were identified from the Sentinel-1a/b DInSAR interferograms using the same method in Section 2.1."

L 318 - What is meant by "might not deflect adequately?". Including a description of how channel is identified in interferograms may help

We have modified this sentence into "Additionally, the narrower ocean trough may lead to ice not fully reaching hydrostatic equilibrium, and therefore bending forces may also influence the response of the ice to ocean tide variations". We have also mentioned the method of identifying ocean channel from interferograms in Line 350-351, please see our last response above.

L 424-425 - Please specify how deep the MUIS GZ is compared to those in Reeh et al, 2000

Good point. The ice thickness at the MUIS western flank GL is about 2500 m according to BedMachine dataset while the glacier thickness at the GL in Reeh et al. (2000) is about 700 m. We have now rephrased this section as "The reason for this tidal phase difference is possibly because the ice at the deep GZ cannot respond adequately in phase with ocean tides (Reeh et al., 2000) at narrow ice shelf embayment such as MUIS. The ice thickness at the MUIS western flank GZ is about 2500 m (Morlighem, 2020; Morlighem et al., 2020) while the glacier ice thickness at the GL in Reeh et al. (2000) is only about 700 m.".

**Technical Corrections**

L 31 - Including the definition of the GL here makes this sentence a bit awkward; consider revising

Agree. We have moved the definition of the GL to the end of this paragraph: "As a result, TGIS and MUIS have been undergoing high basal melting compared with other regions in East Antarctica (Adusumilli et al., 2020; Depoorter et al., 2013; Pritchard et al., 2012), including at their deep grounding lines (GLs) (Chuter and Bamber, 2015; Morlighem et al., 2020) - the location where the grounded ice first comes into contact with the ocean and becomes afloat."

L 44 - "Despite the importance of this region, this is the only study…" - which study is "this" referring to? Li et al., 2016?

This study refers to Li et al. (2015). We have modified this sentence to "Despite the importance of this region, Li et al. (2015) has been the only study on GL migration of Totten Glacier from satellite observations due to the lack of available satellite data and limited spatial coverage."

L 53 - Would be helpful to reference the specific figure in Fricker & Padman, 2006 or include diagram labeling GL proxies in supplement

Agree. We have now referenced the Figure 1 in Fricker and Padman (2006).

Fig. 1 - For consistency with later figures, consider using same color for DInSAR 1996 GL

We kept the color of MEaSUREs GLs in Figure 1 as black because the GLs used in this plot contain all the GL acquisitions available from multiple years across the Wilkes Land. However, we have revised the GL colors in all the other DInSAR GL plots to be consistent throughout the manuscript. Please also see our previous responses.

L 80 - sentence is a bit awkward

Agree, we have modified this sentence to "The single look complex (SLC) synthetic aperture radar (SAR) images in wide swath mode from both Sentinel-1a/b satellites with a 6-day repeat cycle were used to construct DInSAR interferograms between July 2017 and September 2021. These interferograms were used to derive the GL locations of the Moscow University and Totten Glacier ice shelves in Wilkes Land."

L 82 - include DEM acronym definition

Agree and done.

L 82-83 - "We differenced the two interferograms…" which two interferograms are you referring to?

We have rephrased this sentence into "The DInSAR was generated by differencing two consecutive 6-day SAR interferograms. This differencing removes signals such as time-invariant velocity, to identify ice flow deformation signals caused by ocean tides at the GZ."

L 89 - include the number of usable interferograms for each of MUIS West, TGIS East channel and TGIS east branch ice stream, and reference boxes from Fig. 1. Please consider including a table or adding to Table 1 the number and nature of derived grounding line positions for each location (e.g. # of interferograms, whether GL position is a single point as in case of ICESat/-2 or a curve as in case of interferograms)

Good advice. We have added the number of usable interferograms and reference boxes from Figure 1b for Totten Glacier East Branch (TGEB), Moscow University Western Ice Shelf (MUWIS) and Totten Glacier Eastern Ice Shelf (TGEIS) in the main text and included the information in Table 1:

Line 93-95: However, we were able to map the GLs at Totten Glacier East Branch (TGEB, box B in Figure 1b), Moscow University Western Ice Shelf (MUWIS, box D in Figure 1b) and TGEIS (box C in Figure 1b), the numbers of usable DInSAR interferograms in each region are 11, 7 and 14, respectively.

Table 1. The mean grounding line (GL) migration distance and standard deviation in five studied regions including Moscow University Ice Shelf (MUIS) eastern flank, MUIS western flank, Totten Glacier Ice Shelf (TGIS) southern lobe, Totten Glacier East Branch (TGEB), and Moscow University Western Ice Shelf (MUWIS).

| Region | Time Period | Instrument | GL Feature Type | Number of GL Measurements | Mean GL Migration (km) | Standard Deviation (km) |
|---|---|---|---|---|---|---|
| MUIS eastern flank | 1996-2020 | ICESat-2 | Point | 8 | 9.37 | 1.04 |
| MUIS western flank | 1996-2020 | ICESat-2 | Point | 2 | 13.85 | 0.08 |
| TGIS southern lobe | 1996-2020 | ICESat-2 | Point | 12 | 3.51 | 0.49 |
| TGEB | 1996-2020 | Sentinel-1a/b | Line segment | 11 | 5.95 | 0.75 |
| MUWIS | 1996-2021 | Sentinel-1a/b | Line Segment | 1 | 11.95* | - |

*Note the GL migration distance for the MUWIS is not mean value as there is only one present-day Sentinel-1a/b DInSAR interferogram available.

L 101 - Include track number(s) here

Agree and done.

L 104 & 107 - are the saturation correction and tide corrections included with GLAS data? Please include reference

Yes, the saturation correction and tide corrections are all included in the GLAS data. We added the following reference:

Zwally, H. J., Schutz, R., Dimarzio, J. and Hancock, D.: GLAS/ICESat L2 Global Antarctic and Greenland Ice Sheet Altimetry Data (HDF5), Version 34, Natl. Snow Ice Data Cent. Distrib. Act. Arch. Cent., doi:https://doi.org/10.5067/ICESAT/GLAS/DATA209, 2014.

L 114 - Consider including ICESat ground track on Fig. 2

Agree and done.

L 126 - is ocean-loading tide included with ATL06 data? Please include reference

Yes, the ocean-loading tide is included in the ATL06 data, we added the following reference:

Smith, B., Fricker, H. A., Gardner, A., Siegfried, M. R., Adusumilli, S., Csathó, B. M., Holschuh, N., Nilsson, J., Paolo, F. S. and and the ICESat-2 Science Team: ATLAS/ICESat-2 L3A Land Ice Height, Version 4, NASA Natl. Snow Ice Data Cent. Distrib. Act. Arch. Cent., doi:https://doi.org/10.5067/ATLAS/ATL06.004, 2021.

L 132 - please ensure that supplemental and main-text figures are referenced in order in the text

Agree and done.

L 135 - why is "elevation anomalies" in quotes? Is it defined differently from in the previous section?

The definition of the elevation anomalies is the same with the previous section. We have removed the quotes here.

L 153 - should section title read "CryoSat-2 elevation change rates…" ?

Agree and done.

L 170-175 - Please define h_f in the text

Agree and done.

L 202 - absolute tidal range is defined as |dh| here but |δh| in figures - please review for consistency

Agree. We have now changed all the dh to δh throughout the manuscript.

Fig. 2 - Mention flowline in caption and consider placing subplots b-g in chronological order. Please also consider differentiating the DInSAR GLs in (a) by dates from (b-g) by color/symbol/etc for easier interpretation

Agree and done. The modified Figure 2 is shown below, the reason of keeping DInSAR GLs in subplots b-g is because the black solid line can better stand out from the background rainbow colormap of interferograms, please also see our previous responses on this point.

[Figure]

Figure 2. a) Grounding line (GL) distributions at the main glacier trunk of Moscow University Ice Shelf (MUIS) overlaid with the ice surface velocity magnitudes and ice flow directions (white arrows) (Rignot et al., 2017). The ICESat-2-derived inland limit of tidal flexure (Point F) locations are shown as maroon dots, and the ICESat-derived Points F locations from Figure S1 are shown as orange dots. b-g) Sentinel-1a/b DInSAR interferograms of MUIS between 2018 and 2021, the ice flowline is shown as the white solid line. In all subplots, the 1996 MEaSUREs DInSAR GLs (Rignot et al., 2011, 2016) and the 2018 DInSAR GLs (Mohajerani et al., 2021) are shown as blue solid lines and green solid lines, respectively. The GLs delineated from Sentinel-1a/b DInSAR interferograms in our study are shown as yellow (2018) and brown (2021) solid lines in subplot a, and black solid lines in subplots b-g.

Fig. 4 - Please specify in the caption that the zero mean tidal amplitudes from CATS2008b are on the bottom panel of each plot. It is also unclear why for 4 a & d the "manually defined reference GL" (is this defined elsewhere?) is specified and for b & c the "DInSAR ice shelf" is specified. Please consider placing subplots in order from furthest upstream to furthest downstream (or vice versa)

Thank you for the advice. We have now specified in the caption that the zero mean tidal amplitudes from CATS2008 tidal model are on the bottom panel of each plot. We manually defined a reference GL upstream of the 1996 MEaSUREs GL for MUIS is to make sure no GL retreat is omitted in our ICESat-2 GZ calculation caused by the uncertainty in the predefined reference GL (mentioned in Section 2.3), and this reference GL is shown as the white dashed line in Figure S2a. The reason of using the "manually defined reference GL" in subplots 4a, d instead of the "1996 DInSAR ice shelf" because there is no intersection between 1996 DInSAR GL and the Track 1108 GT2R (Fig. 4a) and the track 841 GT1L (Fig. 4d) as shown from Figure 2a. We think the order of subplots here is not very important as the subplots are constructed and labelled by the track numbers which make them very easy to refer to.

L 250 - Mention which zoom box in Fig. 1 corresponds to the TGIS southern lobe

Agree and done.

L 273 - Mention which zoom box in Fig. 1 corresponds to Totten East branch

Agree and done.

Fig. 6/L 285 - the caption indicates that the Mohajerani GL is shown in all plots, but it is only shown in 6a. Please revise.

Agree and done.

L 313 - Mention which zoom box in Fig. 1 corresponds to MU western ice shelf

Agree and done.

L 368 - Please define "zero mean annual ice velocity changes"

We changed this sentence into "Zero mean annual ice velocity which was calculated as the deviation from the mean ice velocity...".

L 395-396 - "...maintainted stability"?

We have now changed "maintained stable" to "remained almost constant".

L 414-419 - Consider revising for clarity by describing the effort of Konrad et al., 2018 (L 417-418) before comparing it to your results, and please check sig figs

We have rephrased these sentences into "The GL retreat rates based on ICESat-2-derived Point F locations and the 1996 MEaSUREs GL are 0.36 – 0.46 km yr$^{-1}$ along the ice flowlines at the eastern flank of MUIS (Table S3). This contradicts the highest retreat rate of 0.079 ± 0.029 km yr$^{-1}$ in this region calculated by Konrad et al. (2018), which derived the GL migration rate based on a hydrostatic equilibrium assumption using CryoSat-2-derived surface elevation and the Bedmap-2 bed elevation at the MEaSUREs GL locations (Rignot et al., 2011) instead of mapping the GL location directly. However, on the western flank of MUIS, the GL retreat rate from ICESat-2 between 1996 and 2020 is around 0.57 km yr$^{-1}$, at a similar magnitude to the highest retreat rate of 0.25 ± 0.099 km yr$^{-1}$ observed by Konrad et al. (2018). The bed elevation in MUIS GZ has large uncertainty especially on MUIS due to the lack of airborne radar data coverage (Fig. S32 in Morlighem et al. (2020)), where the difference between BedMachine and Bedmap-2 bed elevations can be up to 1 km. Therefore, the approach used in Konrad et al. (2018) is likely to have large uncertainties in regions like MUIS, which might cause discrepancies with our GL retreat rates."

L 479-481 - Please specify the time periods over which retreat was observed and which metric of retreat is used (maximum, mean, median?)

Agree, we rephrased this sentence to "The mean GL retreat distances in 1996-2020 at the eastern and western flanks of Moscow University Glacier's main glacier trunk are 9.37 ± 1.04 km and 13.85 ± 0.08 km, respectively. Meanwhile during the same period, the mean GL retreat distances are 5.95 ± 0.75 km and 3.51 ± 0.49 km at TGEB and the southern lobe of the TGIS main glacier trunk, respectively."

L 485-486 - Please revise "...interferogram time series also discovers..." - I suggest "...time series allowed us to discover.." or "..time series revealed..."

Agree, we changed this sentence to "The analysis of Sentinel-1a/b DInSAR interferogram time series revealed a tide-modulated ocean channel located at MUWIS."

L 493 - "..unable to identify the exact time stamps of GL retreats" - you do identify retreat over as specific as 6-day periods and successfully link it to tidal motion, so the reason for this phrasing is unclear to me. Please clarify.

Agree, we have rephrased this sentence to "Although we are unable to identify the exact time stamps when GLs started retreating, the findings offer new insights into the current GL and ice sheet instability in Wilkes Land".

L 503 - typo in ESA url

Agree, we corrected the ESA url to https://science-pds.cryosat.esa.int/.

**Major Review Comments by Bert Wouters and Rob Bingham**

1. For the revised TC paper, decide on a consistent way of naming the western MUIS feature.

Agree and done. I have revised the naming of each studied region in Chapter 6 as follows, modified the naming in all the relevant figures and updated the acronyms to the *List of Abbreviations*.

- TGIS: Totten Glacier Ice Shelf
- TGEB: Totten Glacier East Branch
- TGEIS: Totten Glacier Eastern Ice Shelf
- MUIS: Moscow University Ice Shelf
- MUWIS: Moscow University Western Ice Shelf

2. Everywhere that you state correlation values, please include the p-value.

Thanks for this comment. We have now calculated the p-value for each correlation map of GL migration and tidal range, they are denoted on the figures and mentioned in the associated text. Note the p-value of Figure 3 is 0.2, this is caused by the limited 3 data samples we have, and we mentioned this in Line 221: "A high positive correlation is found between the GL migration distance and the differential tidal amplitude $\delta h$ with an R-squared value of 0.91 (p-value > 0.1 given we only have three samples) (Fig. 3)"

3. Include plots with GL migration vs non-absolute tidal range, and revise associated text accordingly.

We added the non-absolute differential tidal amplitudes σh for the Moscow University Ice Shelf (MUIS) (Fig. 3) and Totten Glacier East Branch (TGEB) (Fig. 7) as the tidal change direction will have an impact on the grounding line migration direction (negative σh results in advance while positive for retreat in relation to the 1996 GL).

However, I kept the absolute differential tidal amplitudes σh for the two ocean channels identified at Totten Glacier Eastern Ice Shelf (TGEIS) and Moscow University Western Ice Shelf (MUWIS) due to the reason that any changes in differential tidal amplitudes (either positive or negative) will only result in the expanding of the ocean channel – this means the channel width will always be positive compared with the zero channel width in 1996. The σh for each DInSAR interferogram is calculated as the difference between two InSAR interferograms formed at three different time stamps: σh = (h3-h2) − (h2-h1). Therefore, DInSAR measures the deformation across the time period between t1 and t3. If σh = 0, it means there is no tidal change happening during this period, the channel width caused by tidal deformation should be zero as a result. If σh >0, it means the tidal amplitude has been increasing through the DInSAR period, the channel should be expanding, and the channel width should be increasing compared with the baseline (when the channel width is zero). However, if σh < 0, although it indicates that the tidal amplitude has been decreasing and channel should be shrinking, the channel width is still positive as the channel is still open at this stage. The point we wanted to make through these two figures, is that tidal variations can result in a change in channel width, the higher the differential tidal amplitude magnitude, the wider the ocean channel.

4. For the overall structure of the revised TC paper, try to be consistent in the narrative order of discussing TGIS and MUIS. In the current writing, this seems rather random.

Agree and done. We have re-structured the results and discussion sections in the following order:

- Moscow University Ice Shelf

- Totten Glacier Ice Shelf
- Totten Glacier East Branch
- Moscow University Western Ice Shelf ocean channel
- Totten Glacier Eastern Ice Shelf ocean channel

The subplots of height above HE, bed topography, and ice velocity figures also follow this order.

5. The explanation that a change in ice velocity results from "ice dynamics" needs to be expanded to explain what is causing the change in ice dynamics in the first place.

Agree and done. We modified this sentence to "The iceberg calving at the ice shelf front, and the melt-driven thinning concentrated near the GL and the reduction of basal traction due to GL retreat, can lead to the speedup of grounded ice (Fürst et al., 2016; Gudmundsson et al., 2019; Joughin et al., 2021; Rignot, 2006). This ice velocity acceleration was unlikely .... will not cause major dynamic changes at the GZ. Therefore, a possible explanation for this discrepancy between ice velocity acceleration and small calving events might be melt-driven GL retreat."

**References:**

[revised manuscript text omitted]